# Characterizing the flavodoxin landscape in *Clostridioides difficile*

Daniel Troitzsch,[1] Robert Knop,[1] Silvia Dittmann,[1] Jürgen Bartel,[2] Daniela Zühlke,[1] Timon Alexander Möller,[1] Linda Trän,[1] Thaddäus Echelmeyer,[1] Susanne Sievers[1]

**ABSTRACT** *Clostridioides difficile* infections have become a major challenge in medical facilities. The bacterium is capable of spore formation allowing the survival of antibiotic treatment. Therefore, research on the physiology of *C. difficile* is important for the development of alternative treatment strategies. In this study, we investigated eight putative flavodoxins of *C. difficile* 630. Flavodoxins are small electron transfer proteins of specifically low potential. The unusually high number of flavodoxins in *C. difficile* suggests that they are expressed under different conditions. We determined high transcription levels for several flavodoxins during the exponential growth phase, especially for *floX*. Since flavodoxins are capable of replacing ferredoxins under iron deficiency conditions in other bacteria, we also examined their expression in *C. difficile* under low iron and no iron levels. In particular, the amount of *fldX* increased with decreasing iron concentration and thus could possibly replace ferredoxins. Moreover, we demonstrated that *fldX* is increasingly expressed under different oxidative stress conditions and thus may play an important role in the oxidative stress response. While increased *fldX* expression was detectable at both RNA and protein level, *CD2825* showed increased expression only at mRNA level under $H_2O_2$ stress with sufficient iron availability and may indicate hydroxyl radical-dependent transcription. Although the exact function of the individual flavodoxins in *C. difficile* needs to be further investigated, the present study shows that flavodoxins could play an important role in several physiological processes and under infection-relevant conditions.

**IMPORTANCE** The gram-positive, anaerobic, and spore-forming bacterium *Clostridioides difficile* has become a vast problem in human health care facilities. The antibiotic-associated infection with this intestinal pathogen causes serious and recurrent inflammation of the intestinal epithelium, in many cases with a severe course. To come up with novel targeted therapies against *C. difficile* infections, a more detailed knowledge on the pathogen's physiology is mandatory. Eight putative flavodoxins, an extraordinarily high copy number of this type of small electron transfer proteins, are annotated for *C. difficile*. Flavodoxins are known to be essential electron carriers in other bacteria, for instance, during infection-relevant conditions such as iron limitation and oxidative stress. This work is a first and comprehensive overview on characteristics and expression profiles of the putative flavodoxins in the pathogen *C. difficile*.

**KEYWORDS** *Clostridioides difficile*, flavodoxins, iron limitation, oxidative stress

The term flavodoxin includes a group of small monomeric single-domain flavoproteins, containing one molecule of non-covalently bound flavin mononucleotide (FMN). This cofactor consists of an isoalloxazine ring system which is fused to ribitylphosphate. The FMN molecule lends flavodoxins the ability to catalyze one- and two-electron transfer redox reactions; they are thus also capable of generating or neutralizing radicals. Their vital role in a wide range of metabolic processes as soluble electron transfer proteins has been reported previously (1). Except for some reported eukaryotic green

Address correspondence to Susanne Sievers, susanne.sievers@uni-greifswald.de.

The authors declare no conflict of interest.

See the funding table on p. 22.

10.1128/spectrum.01895-23 **1**

and red algae, flavodoxins are restricted to prokaryotes as stand-alone proteins (2, 3). In higher eukaryotes, the flavodoxin fold is only incorporated into multidomain enzymes, as some oxidoreductases (1).

Flavodoxins have been discovered in the 1960s in cyanobacteria (4) and Clostridia (5) grown under low iron conditions. Knight *et al.* showed that flavodoxins can serve as electron carriers in reactions characteristic for the iron-containing ferredoxins (5). Phylogenetic analyses revealed that both ferredoxins and flavodoxins developed evolutionary early in the anaerobic environment (6). According to redox potentials, flavodoxins largely match those of ferredoxins. The important difference is that flavodoxins use FMN as prosthetic group and not an oxygen ($O_2$)-sensitive [Fe-S] cluster like ferredoxins. Nevertheless, when both electron transport proteins are encoded in the same genome, the oxidation-sensitive metalloproteins often play the key role in the network of electron transfer (7). The isofunctional flavodoxins are typically expressed under environmental stress and nutrition starvation conditions (e.g. iron limitation) (6).

All yet-known flavodoxins are highly acidic proteins of small size (140 to 180 amino acid residues). The typical flavodoxin-like fold is characterized by a three-layered *αβα*-structure with a central five-stranded β-sheet which is surrounded by two α-helical layers (8) (Fig. 1). Flavodoxins can be divided into two groups: short-chain and long-chain flavodoxins. The latter having an approximately 20 amino acid long extra loop of still unknown function within the fifth *β*-strand (1, 9). It is assumed that this extra loop plays an important role in the interaction with other proteins and enhances the affinity of binding the FMN cofactor (9). Cyanobacteria and algae feature exclusively the long-chain form, whereas gram-negative bacteria can synthesize flavodoxins of either category. Interestingly, in gram-positive bacteria such as *C. difficile,* only short-chain flavodoxins have been reported so far (8).

According to an evolutionary study (10), bacteria of the phylum Firmicutes exhibit an average of one to two flavodoxin genes. Hitherto, there are no published data describing the total number or the importance and function of flavodoxins in *C. difficile* in detail, except for flavodoxin *CD630_19990* (*fldX*), which was shown to be strongly upregulated during iron limitation (11, 12). For several bacteria, like *Escherichia coli* (13) or *Paraburkholderia xenovorans* (14), it has already been reported that flavodoxins are involved in the oxidative stress response. Especially for pathogens, it is important to be able to protect against reactive oxygen species generated by the host's immune system. Previous studies demonstrated *C. difficile* to tolerate oxygen to a greater extent than expected for a strictly anaerobic bacterium (15–18). It could be hypothesized that flavodoxins in *C. difficile* contribute to the surprisingly great aero-tolerance being highly beneficial under infection.

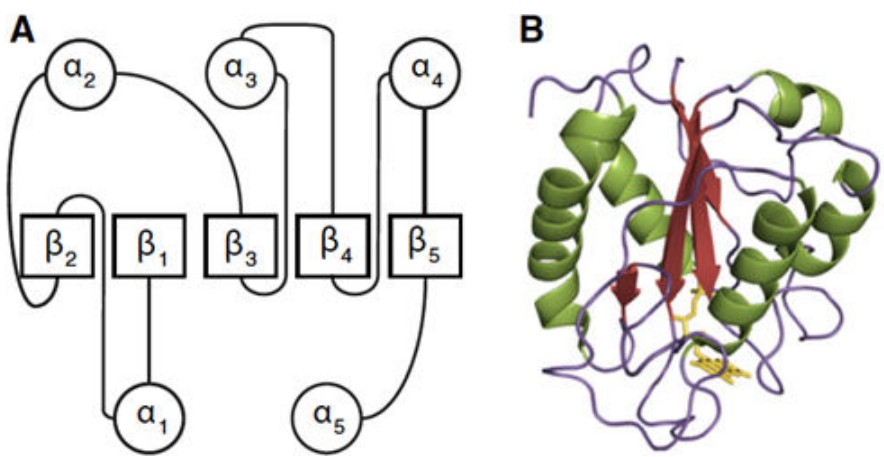

**FIG 1** Flavodoxin fold. Adapted from Houwman and van Mierlo (8). (A) Schematic topology with α-helices as circles and β-strands as squares. (B) An image of flavodoxin II from *Azotobacter vinelandii*. The FMN cofactor is colored in yellow.

In this study, we analyzed all annotated putative flavodoxins of *C. difficile* 630. A comprehensive bioinformatic analysis and an extensive expression profiling provide first evidence on the importance and functional role of the eight flavodoxins. The data also point at putative redundancies and individual functions of specific flavodoxins in *C. difficile,* confirming them as potential targets in novel anti-*C. difficile* therapies.

## RESULTS

### *In silico* analyses

A whole-genome analysis of reference strain *C. difficile* 630 (NCBI Taxonomy ID: 272563) revealed an annotation of eight highly diverse gene sequences encoding flavodoxin-like proteins. According to the Uniprot database (19), the putative flavodoxins have the following gene identifiers: *CD630_08100*, *CD630_14580*, *CD630_16790*, *CD630_19990*, *CD630_22070*, *CD630_26840*, *CD630_28250* and *CD630_31210*. Hereafter, we will use the abbreviated notation of the flavodoxin names, *CD0810 (floX), CD1458 (wrbA), CD1679, CD1999 (fldX), CD2207, CD2684, CD2825,* and *CD3121*. Table 1 lists characteristics specific to each gene and associated protein (Table 1). The eight flavodoxin sequences are distributed over the entire 4,290,252 bp genome of *C. difficile* 630 (20). With an average gene length of 520 bp and a protein mass of 19 kDa, these putative flavodoxins are comparatively small proteins, except for CD3121 being much larger with 28.5 kDa. An alignment of all eight amino acid sequences shows that the proteins are not very similar in sequence. A one-to-one comparison reveals a sequence identity of less than 22 % and a maximum similarity of 39 % (Table S1). Proteins of the flavodoxin class feature a typical three-layered $\alpha\beta\alpha$-sandwich structure (8). Using the programs Porter5 (21) and ColabFold (22), the secondary and tertiary protein structures were analyzed and compared with two known short- and long-chain flavodoxins as reference (short chain: Spfld of *S. pneumoniae*; DVU_2680 of *D. vulgaris* and long-chain: FldA of *E. coli*; Avin_45950 of *A. vinelandii*) (Fig. 2). The predicted structure of *C. difficile* FldX matches the one of typical short-chain flavodoxins. Also, the structures of the putative flavodoxins CD1679, CD2207, and CD2825 resemble the short-chain structure but with one to two additional $\alpha$-helices between the fifth $\beta$-strand and the fifth $\alpha$-helix, protruding from the structure. WrbA rather fits the structure of long-chain flavodoxins, although the typical extra loop, dividing the fifth $\beta$-strand, is shorter compared to the analyzed reference long-chain flavodoxins. Similar to WrbA, FloX features a small extra loop too but according to the secondary structure prediction without dividing the fifth $\beta$-strand. The numbers of $\beta$-strands and $\alpha$-helices in CD2684 suit to that of typical flavodoxins, but the tertiary structure reveals a few discrepancies caused by the much longer sequences between the $\beta$-strands and $\alpha$-helices. In contrast to the topology of the other seven flavodoxins of *C. difficile*, the predicted protein model of CD3121 does not contain a central five-stranded $\beta$-sheet flanked by a pair of parallel $\alpha$-helical layers. Although, CD3121 reveals an N-terminal flavodoxin-like structure, it shows additional $\beta$-strands and $\alpha$-helices after the fifth $\alpha$-helix. This feature is confirmed by an additional "Scan-Prosite" domain analysis (23), revealing that CD3121 features two predicted [4Fe-4S]

**TABLE 1** Characteristics of flavodoxins annotated in *C. difficile* 630

| Flavodoxin | Annotated name | Gene locus | nt-Sequence length (bp) | Read direction | aa-Sequence length (aa) | Protein mass (Da) |
|---|---|---|---|---|---|---|
| *CD0810* | Putative flavodoxin/nitric oxide synthase | 982712–983200 | 489 | Forward | 162 | 17.744 |
| *CD1458* | Putative flavodoxin | 1687897–1688472 | 576 | Forward | 191 | 21.165 |
| *CD1679* | Putative flavodoxin | 1954557–1955036 | 480 | Complement | 159 | 17.785 |
| *CD1999* | Flavodoxin | 2309721–2310149 | 429 | Complement | 142 | 15.611 |
| *CD2207* | Putative flavodoxin | 2557277–2557801 | 525 | Complement | 174 | 20.043 |
| *CD2684* | Putative flavodoxin-related protein | 3103383–3103997 | 615 | Complement | 204 | 23.278 |
| *CD2825* | Putative flavodoxin | 3297715–3298245 | 531 | Complement | 176 | 20.038 |
| *CD3121* | Putative flavodoxin/nitric oxide synthase | 3634567–3635316 | 750 | Forward | 249 | 28.572 |

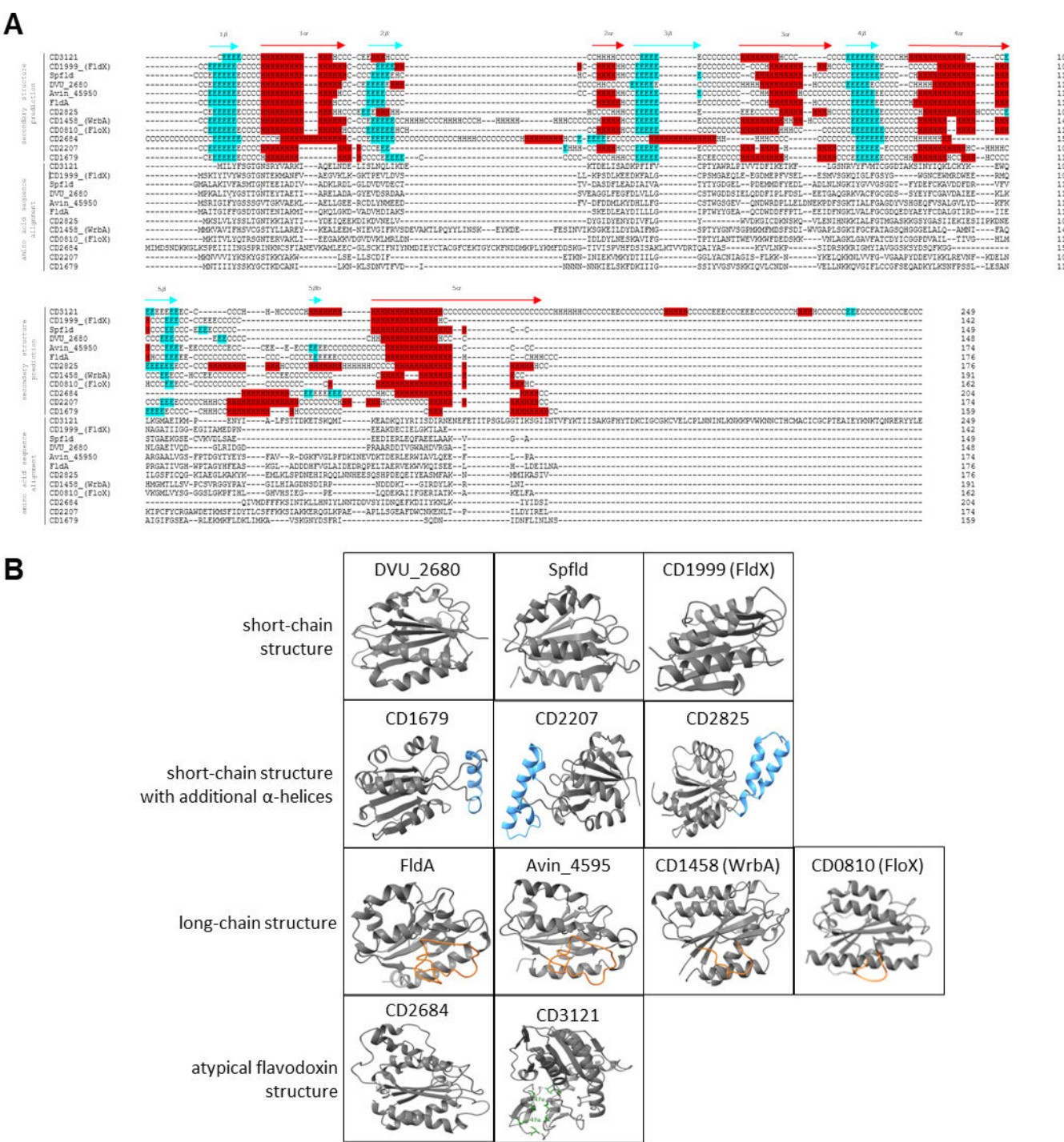

**FIG 2** Comparison of Porter 5-predicted secondary structures and ColabFold-predicted protein structures. (A) The prediction of secondary structure was generated with the three-class secondary structure model of the Porter 5 web tool (H, helix; E, strand; C, coil). Class predictions with a confidence value of at least 5 are highlighted; *α*-helices are red, and *β*-strands are turquoise. Shown are the eight putative flavodoxins [CD0810_(FloX), CD1458_(WrbA), CD1679, CD1999_(FldX), CD2207, CD2607, CD2825, and CD3121) and short-chain flavodoxin SpFld of *Streptococcus pneumoniae* TIGR4, short-chain flavodoxin DVU_2680 of *Desulfovibrio vulgaris*, long-chain flavodoxin FldA of *E. coli,* and long-chain flavodoxin Avin_45950 of *Azotobacter vinelandii*. (B) Comparison of ColabFold-predicted protein structures of eight putative flavodoxins of *C. difficile* 630 and two reference short-chain and long-chain flavodoxins each, pictured with ChimeraX. Distinction features, mentioned in the text are highlighted; extra loops are orange, and additional α-helices are blue. Cysteine residues binding the two 4Fe-4S centers are colored in dark green.

ferredoxin-type binding domains at the C-terminus. We performed a BLAST search of the eight flavodoxin protein sequences to compare them to homologs of other species. For each of the flavodoxins, the 20 species with the highest identity are listed (Table S2). In addition to the expected occurrence of flavodoxins in the *Clostridium* genus, *C. difficile* flavodoxin homologs can be found in quite a large number of other genera. Strikingly, most of the listed organisms have an anaerobic or microaerobic lifestyle. Yet, only one species has been identified having homologs of three of the eight *C. difficile* flavodoxins. *Clostridium estertheticum* encodes homologs of WrbA, CD1679, and CD3121. For WrbA, many homologs are listed as NAD(P)H-dependent oxidoreductases, showing the close affiliation of the flavodoxin sequence to the oxidoreductase protein family. Remarkably, in other species, the sequence of CD3121 seems to be closer to proteins of the ferredoxin family. Altogether, *in silico* results of CD3121 exhibit many discrepancies to a flavodoxin-like fold, and this putative flavodoxin was thus excluded from the following experimental assays. An alignment of the flavodoxin nucleotide sequences of 12 other *C. difficile* strains revealed that all putative flavodoxins have far more than 90 % identity and are thus highly conserved across strains of this human pathogen (Table S3). Figure 3 pictures the genomic localization of the flavodoxin genes in *C. difficile* 630 and their genetic neighborhood (Fig. 3). Additionally, possible operon structures, their transcriptional sizes, and predicted transcription termination sites (TTSs)

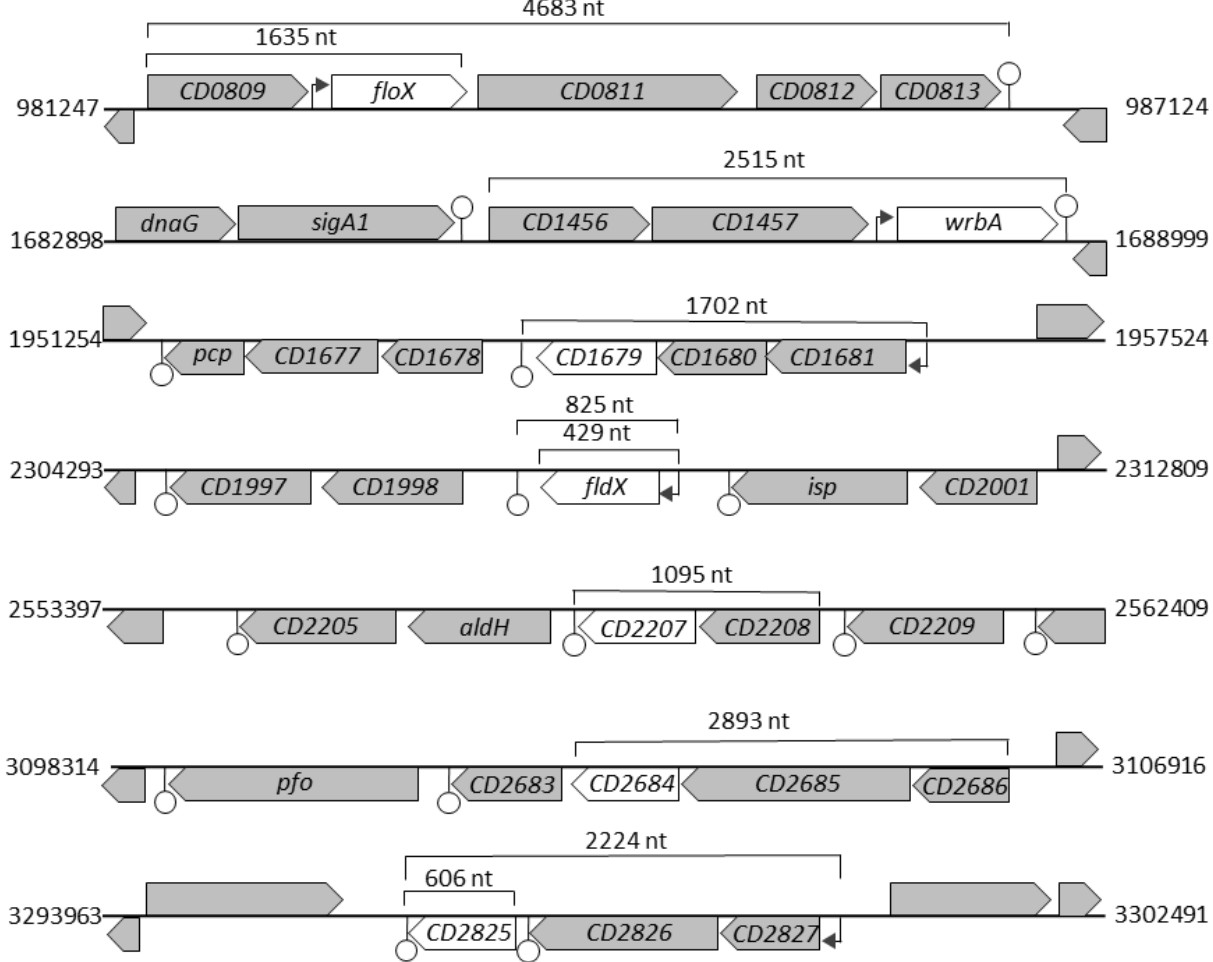

**FIG 3** Genomic neighborhood analysis of the putative flavodoxins *CD0810* (*floX*), *CD1458* (*wrbA*), *CD1679*, *CD1999* (*fldX*), *CD2207*, *CD2684,* and *CD2825* of *C. difficile* 630. The transcriptional start (gray arrow) and terminator sites (circle) were added if they are annotated in the transcription maps of Soutourina *et al.* and Fuchs *et al.* (24, 25). Transcription termination sites were additionally added if predicted by ARNold web service (University Paris-Sud). Shown nucleotide (nt) sizes are potential transcripts according to our Northern blot analysis.

are shown, which were calculated by various software tools and complemented by the transcriptional map generated by Soutourina *et al.* and Fuchs *et al.* (24, 25). The Operon-mapper web server (26) combines the intergenetic distance of the neighboring genes and their associated functional relationship for operon prediction. These *in silico* results revealed a monocistronic reading frame for *CD0810* (*floX*) and *CD1999* (*fldX*). For only one putative flavodoxin, a bicistronic transcript is calculated, i.e., *CD2207* and a downstream neighbored transcriptional regulator of the MarR family. For *CD1485 (wrbA)*, *CD1679*, *CD2684*, and *CD2825*, tricistronic operon structures are predicted (Table S4). RNA-seq analysis of Fuchs *et al.* could confirm the tricistronic operon structure of *CD1679* and *CD2825*. Only those transcription units with a detected transcriptional start site were considered in that publication. We aimed at validating these bioinformatics results by an experimental approach.

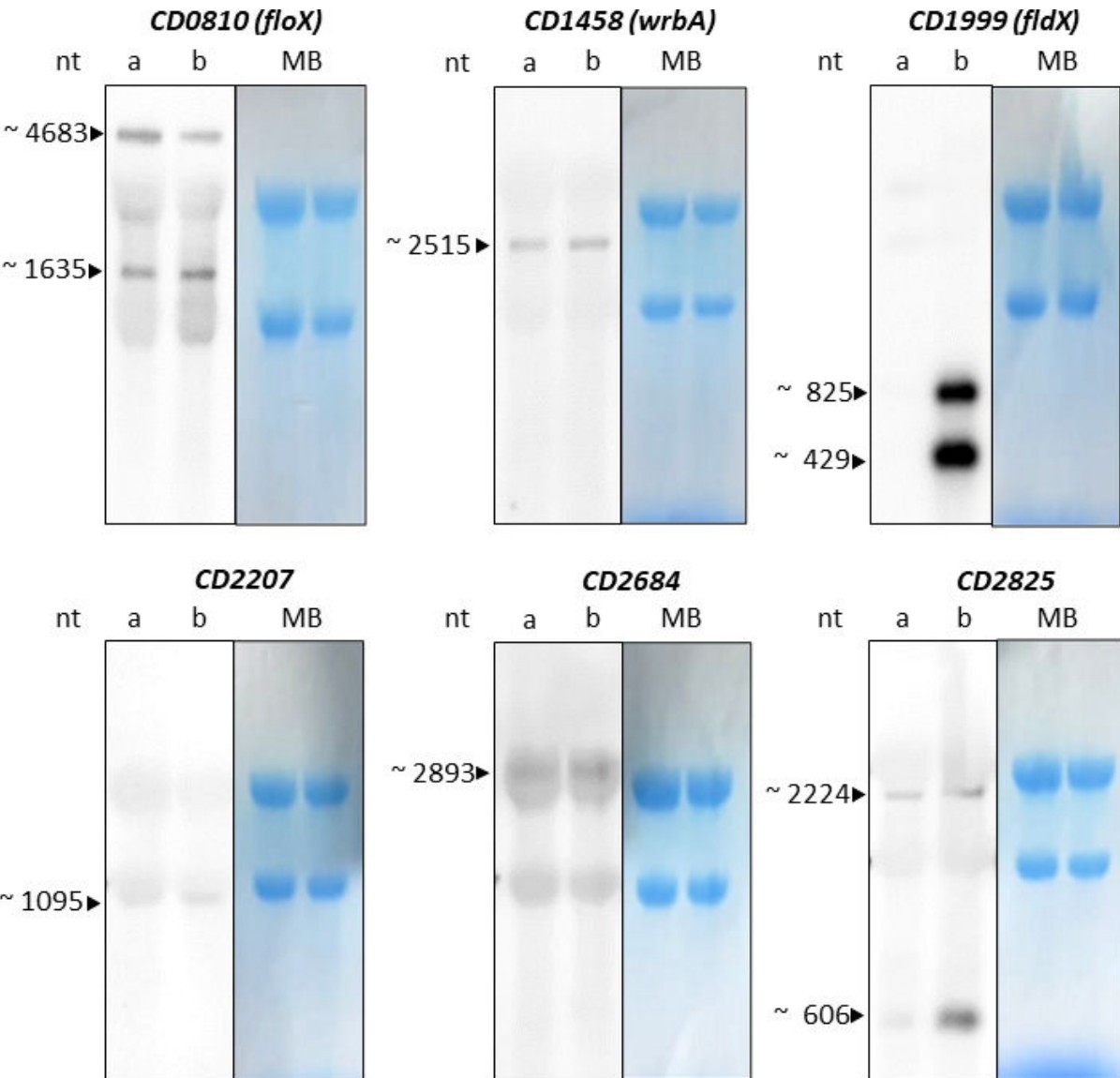

**FIG 4** Northern blot analysis of the expression of the putative flavodoxins *CD0810* (*floX*), *CD1458* (*wrbA*), *CD1999* (*fldX*), *CD2207*, *CD2684*, and *CD2825* of *C. difficile* 630. Lane a, unstressed control conditions; lane b, 10 min 5% oxygen-treated conditions. Molecular sizes were calculated using the DIG-labeled RNA molecular weight marker I (Roche) and compared to the *in silico* operon analysis. Total RNA levels were monitored by methylene blue (MB) staining. nt, nucleotide.

## Experimental and comparative operon analyses via Northern blot

We performed Northern blot analyses of the remaining seven putative flavodoxins to determine transcript length and revise the *in silico* predicted operon structures (Fig. 4). RNA preparations of control and oxygen-treated samples were analyzed. The sizes of flavodoxin mRNAs were compared to a digoxigenin (DIG)-labeled RNA ladder and predicted considering the size of proximate genes (Fig. 4). For *CD1679,* no Northern blot is shown, as there was no signal detectable under the tested conditions. Only *fldX* and *CD2825* showed a band corresponding to their annotated gene size assuming their monocistronic expression. The remaining putative flavodoxins revealed signals with much higher nucleotide sizes than their predicted gene length. These observations indicate a polycistronic operon structure for *floX*, *wrbA*, *CD2207,* and *CD2684*. The experimental results correspond to the *in silico* predicted operon structures (Fig. 3), with the exception of *floX*, which was predicted to be a stand-alone gene. Remarkably, for *floX*, *fldX*, and *CD2825,* two distinct bands could be observed, suggesting a potential read through or more than one promoter or terminator for these genes. Regarding the gene neighborhood of *floX*, the 4,683-nt transcript size could be explained, if all genes are transcribed to the predicted TTS downstream of *CD0813*. The 1,635-nt long smaller transcript fits the size of the gene pair *CD0809* and *floX*, even though there is no TTS next to *floX* predicted. It should be noted that the *fldX* gene is far off its proximate genes so that a polycistronic operon structure seems unlikely. However, we detected two strong bands under oxygen stress conditions, a smaller one at the size of the appropriate gene length of 429 nt and a larger band with a size between 600 and 1,000 nt (Fig. 4). In accordance with Fuchs *et al*. and the "ARNold Finding Terminators" tool (27), we found a Rho-independent transcription terminator approximately 400 nt further downstream of the gene. This predicted hairpin terminator could be one reason for the second Northern blot signal at a size of 825 nt. For *CD2825,* the two signals could be explained by a stand-alone gene expression and a read through of the TTS in front of *CD2825* yielding a tricistronic transcript. Although this Northern blot analysis should only reveal transcript sizes but no quantitative expression data, it was interesting to see that *fldX* and *CD2825* showed an induction under oxygen stress conditions. Whereas both signals of *fldX* strongly increase in the presence of oxygen, *CD2825* is only induced in the smaller transcript. Thus, under oxidative stress conditions, transcription seems to shift from polycistronic expression to a monocistronic expression of *CD2825*, although an unspecific transcript binding of the RNA probe yielding the higher band cannot be excluded.

## Screening of flavodoxin expression

To unravel the physiological role of the different flavodoxins in *C. difficile*, we screened flavodoxin expression under different growth and stress conditions. Therefore, we cultivated *C. difficile* in *C. difficile* minimal medium (CDMM) with defined amounts of nutrients and harvested cells at different phases of growth [exponential growth phase, transient phase, early stationary phase, late stationary phase (12 h after inoculation), and death phase (24 h after inoculation)]. To profile flavodoxin expression, we analyzed transcriptional levels of the seven flavodoxins using slot blot analyses (Fig. 5). All flavodoxins, except *CD1679,* were detectable in the exponential growth phase. In transient and early stationary phase, *CD2207*, *CD2684,* and *CD2825* had a consistent or slightly decreased number of transcripts compared to exponential growth phase, while transcription of *fldX* decreased significantly in transient and early stationary phase (*fldX*: factor of 0.48 and 0.51). *FloX* showed a significant increase in transcription compared with the exponential growth phase (*floX*: factor of 1.44 and 1.66). For *wrbA*, a significant increase in transcription was not seen until stationary phase (factor of 1.36). In late stationary phase, transcription of all flavodoxins was decreased to a very low level. Remarkably, the transcription of *wrbA*, *fldX,* and *CD2825* increased again in death phase compared to late stationary phase. To ensure technical reliability, 5S rRNA was also

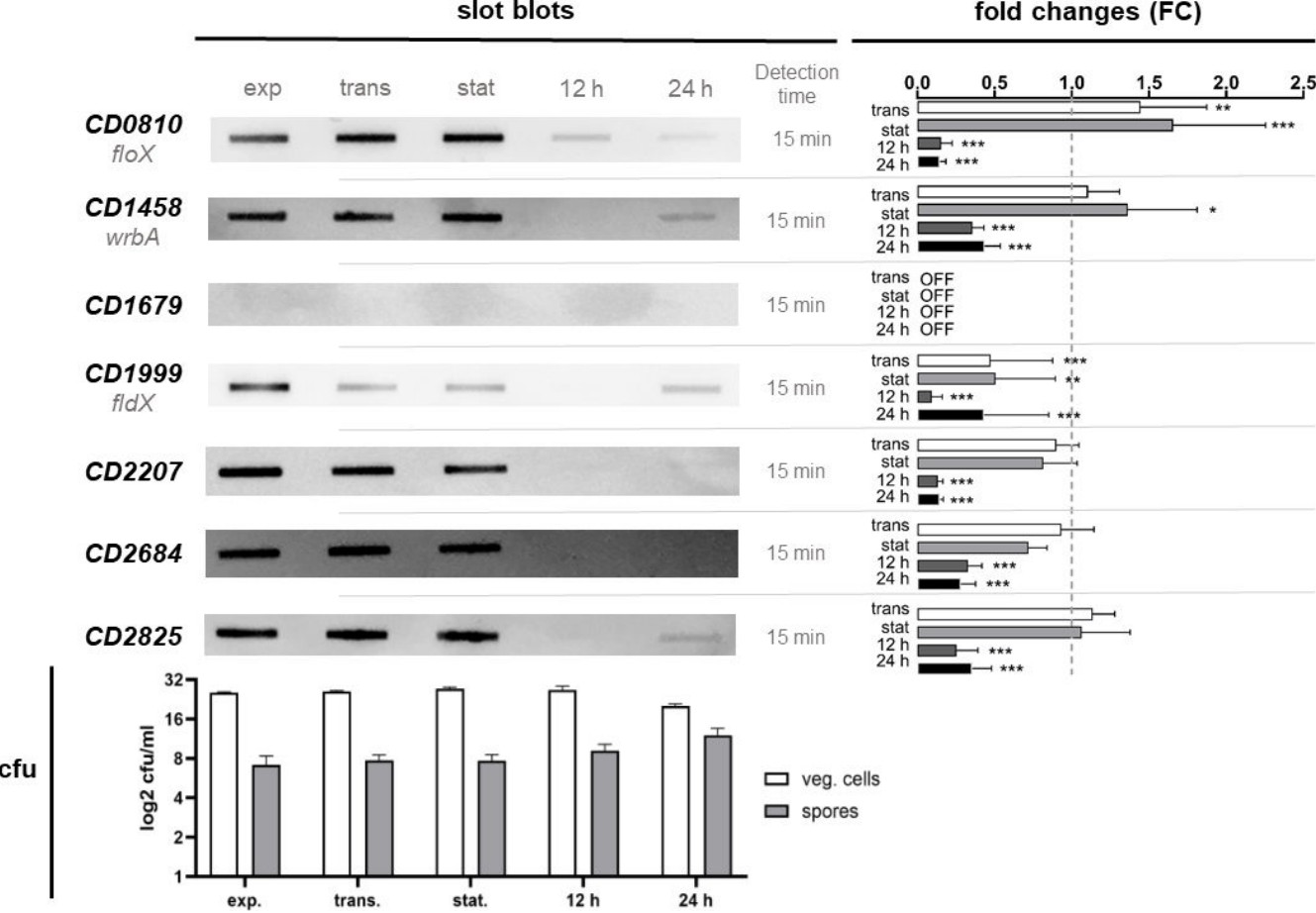

**FIG 5** Expression profile of all known flavodoxins of *C. difficile* 630 during growth. Image of different slot blots, hybridized with the probes named on the left. Out of five replicates per blot, one representative blot of each flavodoxin is shown on the left. The detection time is given next to each blot. Bar charts on the right compare fold changes in relative chemiluminescence signal intensity for each flavodoxin in different growth phases compared to the exponential growth phase. The FC values are plotted as average ($n = 5$) with standard deviation. If signals are completely absent, they are indicated as "OFF." Significant changes determined by a two-way ANOVA for multiple comparisons are indicated ($*P < 0.05$, $**P < 0.01$, $***P < 0.001$). The bar chart in the lower panel shows the logarithmic colony-forming units (cfu) per milliliter of vegetative cells and spores along the growth curve (log2 cfu/mL). The values are plotted as average ($n = 5$) with standard deviation.

detected along the growth curve and stayed constant except for a coherent decreasing signal intensity in death phase (Fig. S1).

In addition, we examined the number of colony-forming units (cfu) and spores in the different growth stages of *C. difficile* to correlate flavodoxin expression with a number of vegetative cells or sporulation. F*ldX* expression increased from 12 to 24 h by a factor of 5 (FC: 12 h: 0.094, 24 h: 0.432). At the same time points, the number of spores increased by 8.5-fold, while the number of vegetative cells decreased by a factor of 136. Although an increase in transcript levels was also detected for *wrbA* and *CD2825* at these time points, the change was much smaller.

Furthermore, we aimed for an expression analysis of flavodoxins on protein level. A shotgun proteomics analysis of extracts from exponential growth phase resulted in an identification of around 1,200 different proteins. However, only flavodoxin FloX was identified in all three replicates with an average amount of 0.022 % of total protein. Therefore, FloX is under the 400 most abundant proteins in *C. difficile* 630 in exponential growth phase (Table S5). For the other six flavodoxins, no reliable protein identification and quantification could be achieved from exponentially growing cells. Due to their low

abundance and small size, a detection of flavodoxins is challenging and requires more sophisticated and targeted mass spectrometry-based approaches.

We continued our flavodoxin expression analyses under different oxidative stress conditions, iron limitation, and a combination of both. In preliminary experiments, we tested the growth of *C. difficile* 630 under different iron concentrations [14.4 µM, 0.2 µM, 0.02 µM, 2 nM, 0.2 nM, and without iron (0 Fe)] (Fig. S2). Since our data demonstrated growth of *C. difficile* 630 even under no iron conditions, we decided to add the iron chelator 2,2' -dipyridyl (DP) in a concentration of 75 or 250 µM as used in various studies before (11, 28, 29). Based on our growth experiments, we continued our study



**FIG 6** Flavodoxin transcription analyses under different oxidative stress conditions and iron concentrations. Slot blots were carried out to investigate the transcription of the seven flavodoxins under different stress conditions. Signals were detected using a FUSION Solo 7S Imaging system and calculated with the Bio1D analysis software. Fold changes (FC) are calculated as average ($n$ = 3). A color code indicates up- (orange) and down-regulation (blue). If signals are completely absent from one of the compared samples, they are indicated as either "ON" or "OFF." Statistical significance was calculated using a multiple $t$-test (*$P < 0.05$, **$P < 0.01$, ***$P < 0.001$). (A) FC was calculated within a defined iron concentration (14.4 µM Fe, 0.2 nM Fe, or 0 Fe) between the oxidatively stressed sample and the corresponding control. (B) FC was calculated between 0.2 nM and 14.4 µM Fe, as well as between 0 and 14.4 µM Fe within a certain oxidative stress condition (control, H$_2$O$_2$, O$_2$, 5 mM Pq, or 5 mM Pq + O$_2$).

with iron concentrations of 14.4 µM, 0.2 nM representing low iron conditions and no supplemented Fe + 75 µM DP representing no iron conditions (0 Fe).

To instigate oxidative stress, we applied 0.4 mM hydrogen peroxide ($H_2O_2$) (16), 5% $O_2$ (17), and 5 mM paraquat (Pq) in combination with 5% $O_2$ for 10 min each. Paraquat is a broad-spectrum herbicide known for the production of superoxide anion ($O_2^-$) in the presence of $O_2$ (30). Results of the flavodoxin transcription analyses under these different stress conditions are shown in Fig. 6. In Fig. 6A, we compared the influence of different oxidative stress stimuli. Therefore, we calculated fold changes of the different oxidative stress conditions against the associated control within a defined iron concentration. In Fig. 6B, we determined the effect of different iron concentrations on flavodoxin expression. We calculated fold changes between 0.2 nM and 14.4 µM Fe, as well as between 0 and 14.4 µM Fe within a certain oxidative stress condition.

A strong induction was visible for *fldX* with highest transcript numbers for $H_2O_2$, $O_2$, and $O_2^-$ stress. *FldX* levels also increased with decreasing iron concentrations without oxidative challenge indicating an induction under both oxidative stress and iron depletion.

*CD2825* also showed a strong reaction to oxidative stress under 14.4 µM Fe with significantly higher transcript numbers. However, a deficiency of iron leads to a decrease in the expression of *CD2825*.

For *wrbA*, a significant induction was detectable in the presence of hydrogen peroxide, whereas *CD2684* transcription was inducible by oxygen treatment. However, these effects decreased with lower iron concentrations and did not reach statistical significance.

For *CD1679*, no transcription was detectable for all tested conditions except for $O_2^-$ treatment under iron depletion, where a slight signal was visible (Fig. S3).

For further quantitative experiments, we focused on flavodoxins *fldX* and *CD2825* because they showed the strongest and significant effects under iron limitation and different oxidative stress stimuli and could thus be the most relevant with respect to infection-related conditions.

## Quantification of *fldX* and *CD2825* expression

To exactly quantify *fldX* and *CD2825* expression under iron limitation and oxidative stress conditions, we used RT-qPCR to determine differences in transcription (Fig. 7 and 8). The used stress conditions were the same as in slot blot analyses. In Fig. 7A and 8A,

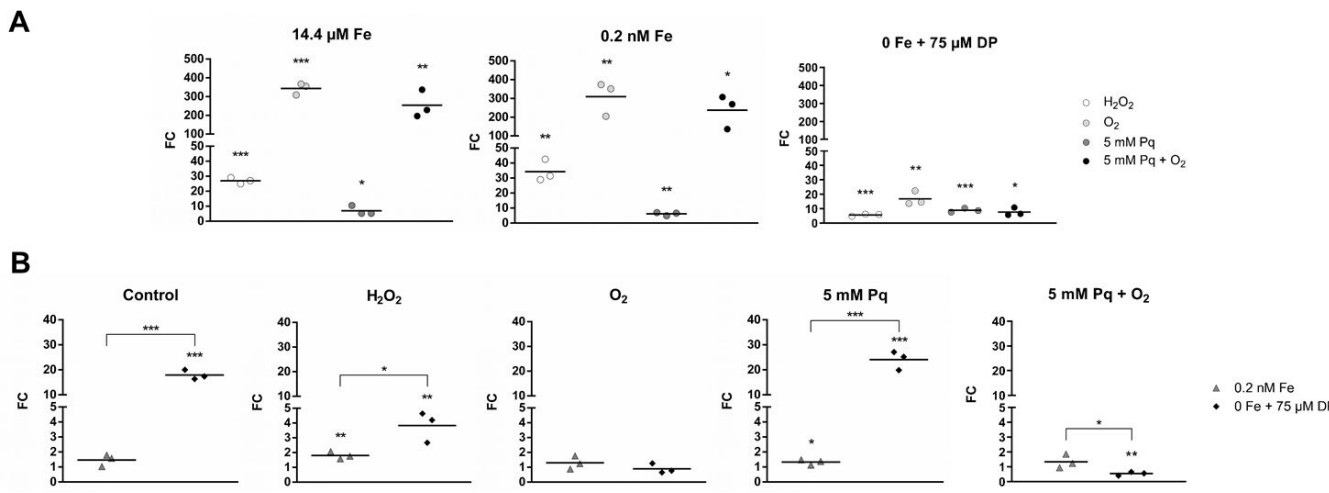

**FIG 7** Transcription of *fldX*. Transcription of the *fldX* gene was quantified for *C. difficile* 630 by RT-qPCR analysis. Statistical significance was calculated using a multiple *t*-test. *, $P < 0.05$; **, $P < 0.01$; ***, $P < 0.001$. The line in the graphs shows the mean. (A) Fold changes (FC) were calculated within a defined iron concentration (14.4 µM Fe, 0.2 nM Fe, or 0 Fe) between the oxidatively stressed sample and the corresponding control. (B) FC were calculated between 0.2 nM and 14.4 µM Fe, as well as between 0 and 14.4 µM Fe within a certain oxidative stress condition (control, $H_2O_2$, $O_2$, 5 mM Pq, or 5 mM Pq + $O_2$).

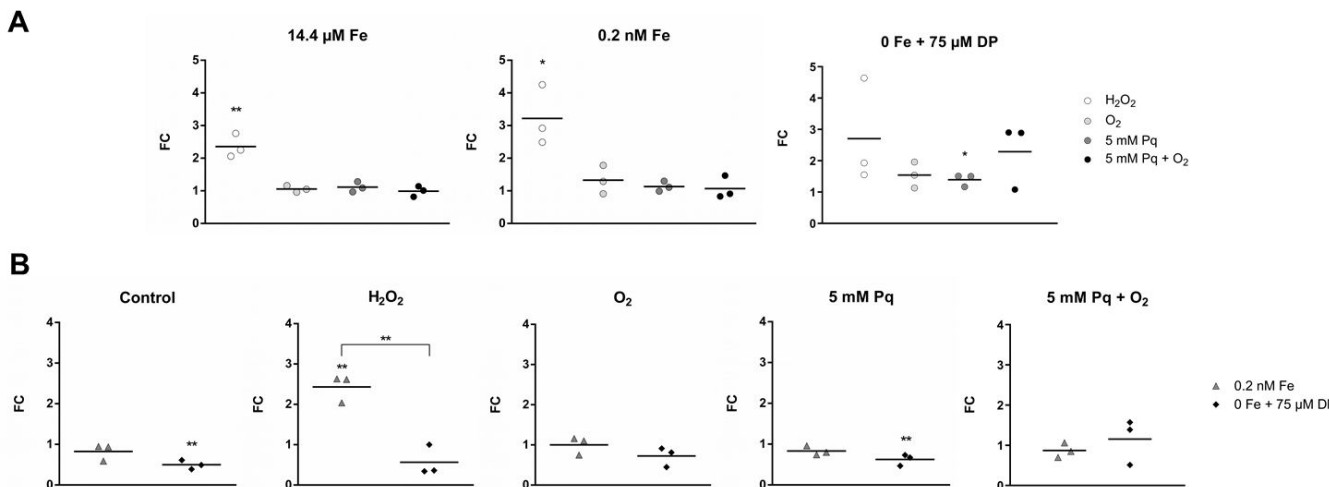

**FIG 8** Transcription of *CD2825*. Transcription of the *CD2825* gene was quantified for *C. difficile* 630 by RT-qPCR analysis. Statistical significance was calculated using a multiple *t*-test. *, $P < 0.05$; **, $P < 0.01$; ***, $P < 0.001$. The line in the graphs shows the mean. (A) Fold changes (FC) were calculated within a defined iron concentration (14.4 µM Fe, 0.2 nM Fe, or 0 Fe) between the oxidatively stressed sample and the corresponding control. (B) FC were calculated between 0.2 nM and 14.4 µM Fe, as well as between 0 and 14.4 µM Fe within a certain oxidative stress condition (control, $H_2O_2$, $O_2$, 5 mM Pq, or 5 mM Pq + $O_2$).

we compared the influence of different oxidative stress conditions within a defined iron concentration, whereas Fig. 7B and 8B show changes in flavodoxin expression under different iron concentrations keeping a certain oxidative stress condition constant.

Comparable to slot blot analysis results, *fldX* showed a strong induction under oxidative stress conditions. Under full iron conditions, the number of transcripts increased drastically by a factor of ~340 for $O_2$ stress and ~250 for $O_2^-$ stress compared to control conditions. A smaller effect was visible under $H_2O_2$ stress (factor of ~27). A similarly strong increase of *fldX* levels was detected under low iron conditions. In the absence of iron, *fldX* transcription is only slightly induced for all oxidative agents (Fig. 7A).

When control samples (no oxidative stress) were compared between different iron concentrations, a sharp increase in *fldX* expression was evident under iron exclusion. Whereas the control and paraquat control showed an expression level of ~1 under low iron conditions, the number of *fldX* transcripts significantly increased under iron exclusion compared to full iron concentrations (control: factor of 17.92; Pq control: factor of 24.08). *FldX* expression was also induced in $H_2O_2$ stressed cells comparing full iron and iron exclusion conditions, although the effect is not as strong as in the control and paraquat control due to the already increased *fldX* expression in the presence of $H_2O_2$. Because during $O_2$ and $O_2^-$ stress in fully iron-supplemented medium *fldX* already showed very high expression levels, no further induction of *fldX* could be detected with decreasing iron concentration under these oxidative stress conditions (Fig. 7B).

The results of RT-qPCR of flavodoxin *CD2825* do not entirely correspond to the results of the initial slot blot analysis (Fig. 8). The quantification showed a significant induction under $H_2O_2$ stress with 14.4 µM (2.3-fold) and 0.2 nM (3.2-fold) iron. For all other tested conditions, the *CD2825* levels show no significant differences.

## FldX and CD2825 protein quantification via parallel reaction monitoring

To validate FldX and CD2825 regulation also on protein level, we used a targeted mass spectrometry approach, i.e., parallel reaction monitoring (PRM). The same oxidative stress conditions and iron concentrations as chosen for RT-qPCR analyses were applied (Fig. 9 and 10).

For FldX, results of the protein quantification are similar to the quantification on transcriptional level. In full iron medium, a significant increase of FldX under $O_2$ and $O_2^-$ stress was detected compared to the respective non-stressed control ($O_2$: factor of ~235, $O_2^-$: factor of ~145) (Fig. 9A). $H_2O_2$ treatment also resulted in increased protein levels

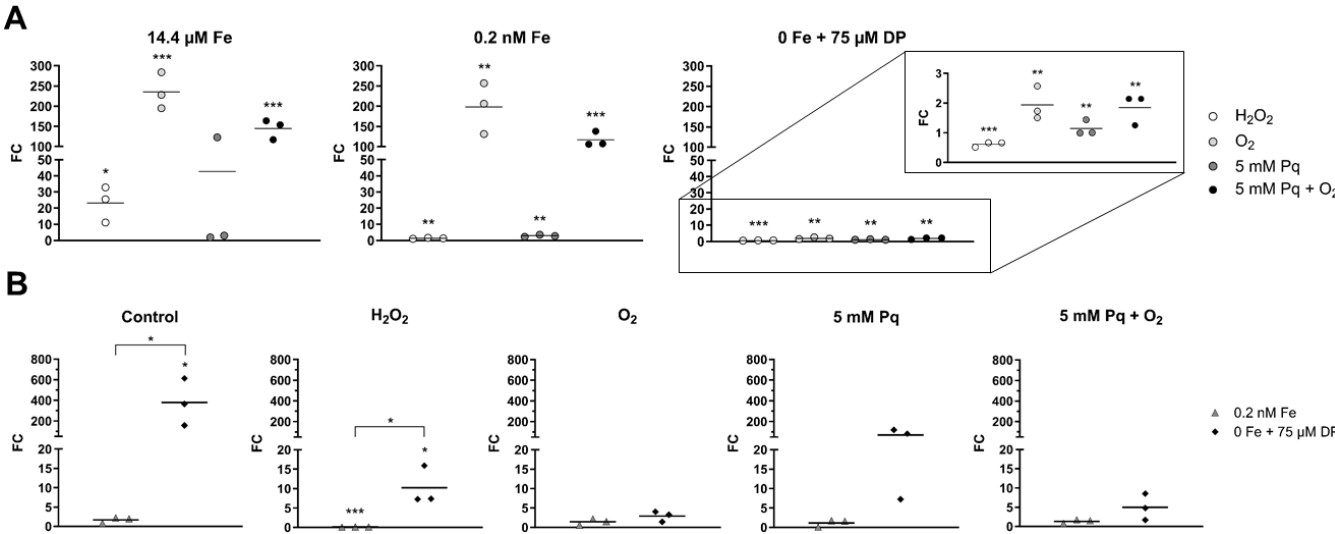

**FIG 9** Protein quantification of FldX. FldX expression on protein level was quantified from cells grown under varying iron concentrations and stress conditions by LC-ESI-MS/MS using parallel reaction monitoring. Statistical significance was calculated using a multiple *t*-test. *, $P < 0.05$; **, $P < 0.01$; ***, $P < 0.001$. The line in the graphs shows the mean. (A) Fold changes (FC) were calculated within a defined iron concentration (14.4 µM Fe, 0.2 nM Fe, or 0 Fe) between the oxidatively stressed sample and the corresponding control. (B) FC were calculated between 0.2 nM and 14.4 µM Fe, as well as between 0 and 14.4 µM Fe within a certain oxidative stress condition (control, $H_2O_2$, $O_2$, 5 mM Pq, or 5 mM Pq + $O_2$).

(factor of ~23), although not as high as the other oxidative stress conditions. Protein amounts of FldX at low iron concentration were similar to 14.4 µM iron conditions. When iron was depleted, there was almost no further but statistically significant increase in FldX abundance when additional oxidative challenge was applied (factor of 0.65 to 2). Compared to cultivation with 14.4 µM and 0.2 nM Fe, the abundance of FldX increased by a factor of ~380 when iron was depleted with DP. Also, in a $H_2O_2$ stress background, an increase of FldX was observed under iron exclusion since $H_2O_2$ alone does not induce gene expression as much as $O_2$ and Pq + $O_2$ do, leaving room for an additional induction

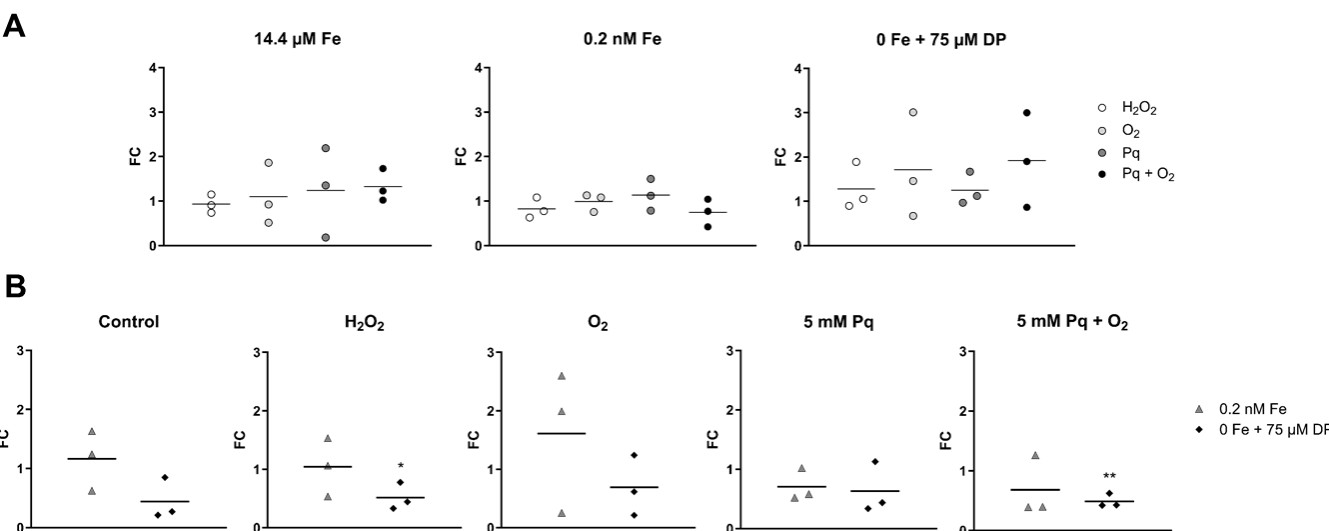

**FIG 10** Protein quantification of CD2825. CD2825 expression on protein level was quantified from cells grown under varying iron concentrations and stress conditions by LC-ESI-MS/MS using parallel reaction monitoring. Statistical significance was calculated using a multiple *t*-test. *, $P < 0.05$; **, $P < 0.01$; ***, $P < 0.001$. The line in the graphs shows the mean. (A) Fold changes (FC) were calculated within a defined iron concentration (14.4 µM Fe, 0.2 nM Fe, or 0 Fe) between the oxidatively stressed sample and the corresponding control. (B) FC were calculated between 0.2 nM and 14.4 µM Fe, as well as between 0 and 14.4 µM Fe within a certain oxidative stress condition (control, $H_2O_2$, $O_2$, 5 mM Pq, or 5 mM Pq + $O_2$).

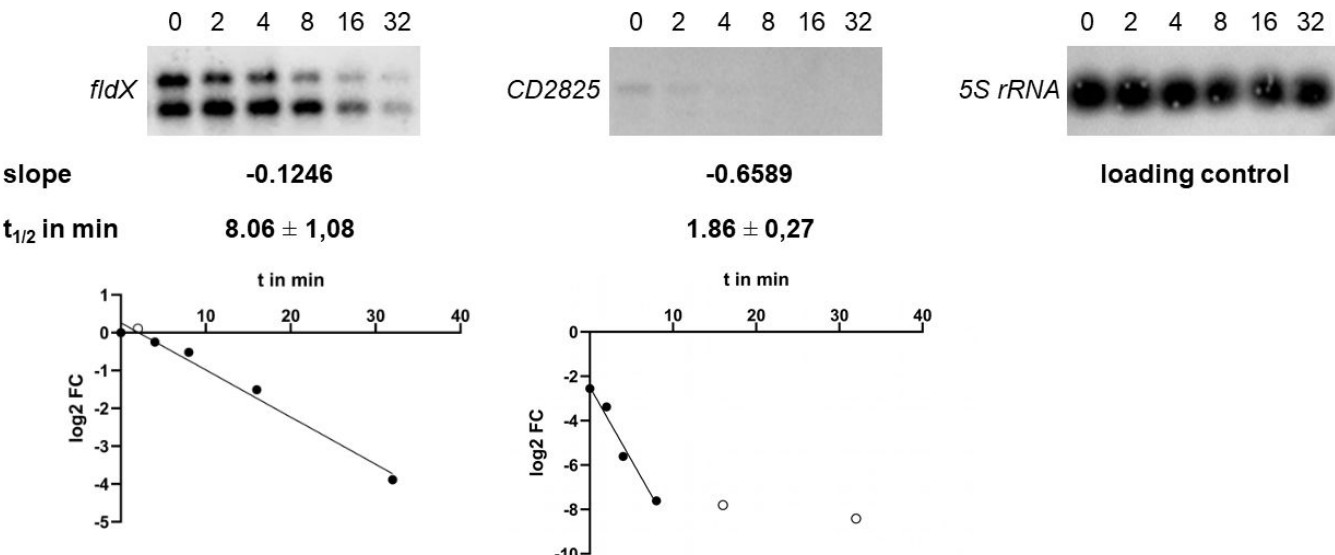

**FIG 11** Determination of transcript half-lives of *fldX* and *CD2825* by Northern blot. The half-lives of *fldX* and *CD2825* were determined after 10 min of $H_2O_2$ stress. The signal of a 5S rRNA-specific probe served as a loading control. Luminescence signals were quantified and normalized against the loading control signals. For the calculation of half-lives, the log ratios of the normalized values (log2 FC) of the respective time points to the normalized value of time point 0 min were formed. These values were plotted against time, and the slope was calculated. The half-life was then calculated from the negative reciprocal of the slope. For the calculation of the slope, only those values were included that were approximately on a straight line (closed circles), those that deviated strongly were not taken into account (open circles). The blots and graphs shown are exemplary of one of three biological replicates.

through iron limitation. A different picture was seen for CD2825. Whereas transcriptional analyses showed an increased expression under $H_2O_2$ stress, the protein amount of CD2825 was approximately the same for all tested conditions (Fig. 10). Thus, results of the PRM measurement do not reflect RNA expression profiles of *CD2825* determined by slot blot and RT-qPCR analyses. Post-transcriptional regulation and mRNA half-life can significantly influence translation of RNA and thus resulting protein levels. Therefore, we determined the half-life of *fldX* and *CD2825* mRNA to estimate their degradation rate.

## mRNA half-life of *fldX* and *CD2825*

To determine the mRNA half-lives of *fldX* and *CD2825*, *C. difficile* was stressed with $H_2O_2$ followed by arrest of transcriptional start by rifampicin addition. Northern blot analyses of *fldX* and *CD2825* transcripts over a time course were performed (Fig. 11). Transcription of 5S rRNA served as a control. The mRNA of *fldX* proved to be fairly stable with a half-life of 8.1 min, in contrast to the mRNA of *CD2825*, whose half-life was determined to be only 1.9 min (Fig. 11).

## DISCUSSION

In this study, we investigated eight putative flavodoxins of *C. difficile* 630. Flavodoxins and ferredoxins, latter ones containing a [4Fe-4S] or [2Fe-2S] iron-sulfur clusters, are small electron transfer proteins of specifically low potential. [4Fe-4S]-ferredoxins are evolutionarily deeply rooted in the genome of ancient organisms that colonize anaerobic environments (10). The [4Fe-4S] cluster is highly sensitive to oxygen. With the advent of oxygenic photosynthesis, the more oxygen-tolerant [2Fe-2S] cluster evolved. However, the increase of oxygen in the atmosphere also led to oxidized and insoluble iron compounds, which reduced the availability of iron for the formation of iron-sulfur clusters. Most likely, this iron limitation led to the formation of flavodoxins, which can replace iron-sulfur clusters by using the cofactor FMN for electron transfer. This theory was supported by a comprehensive analysis of the genomes of algae and cyanobacteria, which revealed a wide distribution of flavodoxin genes in species that colonize iron-poor

environments, but a rarity and absence of flavodoxins in algae and cyanobacteria that live in iron-rich coastal or freshwater environments. This long-term evolution of flavodoxins associated with iron availability and oxygen concentration is reflected in short-term adaptation through gene regulation to rapid changes in iron and oxygen levels (6).

Campbell *et al*. investigated whether genomic endowment with ferredoxins and flavodoxins is associated with the occupied ecological niche of microorganism (10). Strikingly, most of the listed organisms with flavodoxins have an anaerobic or facultative anaerobic lifestyle. Flavodoxins seem to be particularly useful for a survival in the highly iron competitive digestive tract of animals and humans and under highly oxidizing conditions. Campbell *et al*. showed that obligate anaerobic bacteria feature high numbers of ferredoxins with [4Fe-4S] cluster and that the number of ferredoxins with [2Fe-2S] cluster increases with increasing oxygen tolerance. Nevertheless, there was no evidence that the number of flavodoxins in an organism correlates with oxygen tolerance. Furthermore, the number of flavodoxins per genome was investigated in over 7,000 microorganisms. More than six flavodoxins were found in only about 1.55 % of the species examined (e.g., *Methanocella paludicola* SANAE, *Enterobacter* sp. R4-368, *Klebsiella oxytoca* KONIH1) (10).

## The genetic organization of flavodoxins in *C. difficile*

Our genomic analyses revealed eight annotated putative flavodoxins in *C. difficile* 630 indicating an extraordinarily high copy number of these proteins. However, only five flavodoxins were found by Campbell *et al*. in a wide variety of *C. difficile* strains. In addition to the five flavodoxins of Campbell *et al*., we identified CD2684, CD2825, and CD3121 as putative flavodoxins in *C. difficile* 630. An in-depth *in silico* analysis revealed many discrepancies in the structure of CD3121 compared to the typical flavodoxin build. It shows a flavodoxin-like fold at the N-terminus but a C-terminal ferredoxin-like domain. Thus, we considered CD3121 not to be a typical flavodoxin. The ferredoxin-like domain indicates that CD3121 may belong to the family of larger flavodiiron proteins, but a sequence alignment to the known flavodiiron enzymes FdpA and FdpF of *C. difficile* revealed no significant homology (31). In contrast, FldX (CD1999) is a prototype-like flavodoxin showing all structural characteristics of typical short-chain flavodoxins. Also, CD2684 has the typical number and arrangement of *α*-helixes and *β*-strands but with much longer amino acid chains between them. This does not apply to FloX and WrbA, featuring an additional small loop after the fifth *β*-strand very similar to the known long-chain flavodoxins FldA of *E. coli* and Avin_45950 of *A. vinelandii*. This result queries the accepted thesis that gram-positive bacteria exhibit only short-chain flavodoxins (32). Also, the structures of CD1679, CD2207, and CD2825 do not resemble model flavodoxins because of an additional *α*-helix between their fifth *β*-strand and fifth α-helix. In summary, the predicted structures of *C. difficile*'s flavodoxins are various and partially not assignable to the classes of short-chain and long-chain flavodoxins. This classification is thus challenged with the current knowledge and might need to be revised.

Our experimental approach to investigate operon structure corresponds to *in silico* calculated (26) results but in parts differs from annotated operons by Fuchs *et al*. using transcriptional start and termination sites for definition of transcriptional units (24). They predict a possible operon structure only for *CD1679* and *CD2825*, but the other five putative flavodoxins were considered as stand-alone gene. We showed only for *fldX* a monocistronic gene expression but interestingly with two possible transcripts. The strong expression induction seems to lead to a read through of a first TTS to a second one, hypothesizing an occurrence of functional transcript extension for *fldX* under stress conditions. In bacteria, the transcription beyond a terminator sequence is a common mechanism for precise gene regulation (33), e.g., for transcript stability. Furthermore, our Northern blot results revealed two transcriptional units, both being larger than the stand-alone *floX*, contradicting the monocistronic operon structure predicted by both the Operon-mapper and Fuchs *et al*. (24).

## Flavodoxins as electron transfer proteins during exponential growth

We examined the expression of seven flavodoxins of *C. difficile* 630 at different growth stages under standard growth conditions in defined chemical medium via slot blotting. During rapid growth in exponential growth phase, flavodoxins *floX*, *wrbA*, *CD2207*, *CD2684*, and *CD2825* showed high transcription levels, which persisted until early stationary phase. The strongest expression was visible in the slot blot for *floX*, which could also be detected at protein level among the 400 most abundant proteins and as the only flavodoxin that could be detected using a standard proteomics shotgun approach. The exact function and reason for the abundance of FloX in exponential growth phase are subject of future research.

Nevertheless, the results of our study indicate that most flavodoxins in *C. difficile* seem to play an important role as soluble electron transfer proteins in the exponential growth phase characterized by high rates of cell division, growth, and metabolism. Flavodoxins have been studied in a few bacteria, and for several, an essential function has been ascertained. They are especially indispensable in redox reactions involving radical intermediates, as shown for several reactions in *E. coli*, where flavodoxin FldA is involved in the anaerobic activation of pyruvate formate lyase (34), synthesis of methionine (35), activation of class III ribonucleotide reductase (36), and in isoprenoid biosynthesis (37). In *Bacillus subtilis*, both flavodoxins, YkuN and YkuP, were determined to be essential for the synthesis of biotin (38), and in *Helicobacter pylori,* an essential role in the oxidation of pyruvate and reduction of the antibiotic metronidazole was attributed to a flavodoxin (39). In *Azotobacter vinelandii,* flavodoxin 1 participates in nitrate reduction (40). However, none of these flavodoxins shows high homology to one of the flavodoxins in *C. difficile*. Hence, the precise interaction partners of specific flavodoxins in *C. difficile* cannot be inferred from other species and remain unknown but will be investigated in further studies.

As the cell division and growth rate decreased, RNA abundance of flavodoxins also decreased sharply. Presumably, additional flavodoxins are no longer required for electron transfer reactions due to the greatly reduced metabolic activity of cells in the stationary phase.

Remarkably, the transcription of flavodoxins *wrbA*, *fldX,* and *CD2825* increased again in death phase of cells after 24 h. Lysis of dead cells releases nutrients and metabolites. These nutrients could be taken up by living *C. difficile* cells and metabolized for their own survival. The enzymes needed for the recurrent growth might require flavodoxins for electron transfer reactions, which could explain a renewed increase in the expression of flavodoxins. Another explanation could be the onset of sporulation as a strategy for the long-term survival of *C. difficile* during nutrient starvation. Lawley *et al*. analyzed the proteome of the lithium dodecyl sulfate soluble fraction of purified *C. difficile* 630 spores and detected FloX as an electron transfer protein in addition to three ferredoxins (41). Although our data did not show increased transcription of *floX* after 24 h of growth, it could be suggested that some of the flavodoxins are also required for processes involved in sporulation and/or germination. This assumption is supported by the cfu experiments in this study, which show a clear decrease in vegetative cells after 24 h of growth but also a strong increase in spore numbers correlating with the increase of, e.g., *fldX* after 24 h.

## *fldX* is induced by oxidizing agents and iron starvation, whereas *CD2825* expression is affected only by oxidative stress

Flavodoxins and ferredoxins are electron transfer hubs of specifically low potential. Due to the common task spectrum, it seems only logical that both molecules are able to replace each other. In the case of iron deficiency, flavodoxins can replace ferredoxins in their function as electron transfer protein, which can no longer be synthesized due to unavailable [2Fe-2S]-clusters (10). Besides the function to replace ferredoxins, an involvement of flavodoxins in the oxidative stress response has been reported for several bacteria. In *E. coli* both the essential flavodoxin *fldA* and a second flavodoxin *fldB* belong

to the SoxRS regulon controlling the superoxide stress response (13), but their exact roles or details on detoxification reactions within the stress response are still unknown. Also in the strict anaerobe *Clostridium acetobutylicum,* a flavodoxin was increasingly synthesized during $O_2$ challenge (42), and very recently, it was shown that overexpression of two flavodoxins in the aromatic degrader *Paraburkholderia xenovorans* supported survival in the presence of $H_2O_2$ and paraquat (14).

Due to the high number of flavodoxins in *C. difficile*, it is likely that they fulfill different functions and are thus expressed under different conditions. Therefore, we aimed to test all flavodoxins of *C. difficile* for their expression at different iron concentrations and under oxidative stress conditions. We used slot blotting to screen for transcriptional changes. In fact, the strongest induction was detected for *fldX* and *CD2825*. All other tested flavodoxins were either not expressed (*CD1679*) or showed only a slight change in transcription at one or only a few conditions. For example, we detected a significant increase of *wrbA* under $H_2O_2$ stress and of *CD2684* under $O_2$ stress but for both only in fully supplemented iron medium (14.4 µM Fe). A possible contribution of these two flavodoxins cannot be excluded and will be investigated in future studies.

*CD2825* transcription increased especially under $O_2$, $H_2O_2$, and $O_2^-$ stress, whereas *fldX* showed not only an effect under all tested oxidative stress conditions but also with decreasing iron concentrations. *CD2825* and *fldX* have so far not been reported to play a role in the oxidative stress response of *C. difficile*. However, an increased expression of *fldX* in the absence of iron in *C. difficile* is already known (11). Berges *et al.* found *fldX* and genes of riboflavin biosynthesis induced under iron limitation, suggesting a possible substitution of ferredoxin by this flavodoxin (12). In addition, Weiss *et al.* were able to show that, in particular, metal homeostasis genes are upregulated by long-term exposure to low concentrations of oxygen (1.5% for 8 h). Strikingly, *fldX* was with a fold change of 371.9 in strain 630 and 111.5 in strain CD196, the most highly upregulated gene under the tested microaerophilic conditions (43).

To obtain an exact rate of induction of *fldX* and *CD2825*, RNA and protein abundance were quantified by RT-qPCR and PRM analyses, respectively. RT-qPCR results of *fldX* transcription clearly confirm the results of slot blot analyses. Regardless of the amount of supplemented iron, $O_2$ and $O_2^-$ stresses induce a significant increase in *fldX* transcription by more than two orders of magnitude. $H_2O_2$ stress also leads to a significant increase in *fldX* expression. With regard to iron-availability-dependent expression of *fldX*, a clear increase in expression was observed when iron was depleted. The iron concentration in chemically defined medium can be lowered by a factor of 72,000 (14.4 µM to 0.2 nM) having only a small impact on growth (Fig. S2) and without having a major impact on the expression of *fldX*. Growth of *C. difficile* under very low iron concentrations has also been observed in other studies (11, 29). However, our study indicates that the small amount of residual iron seems to be sufficient for the formation of [2Fe-2S]-clusters, thus allowing the formation of ferredoxins. Although affected, *C. difficile* was even able to grow when iron supplementation was completely omitted and chelator 2,2'-dipyridyl added. In this condition, the formation of [2Fe-2S]-cluster seems to be no longer possible, and ferredoxins need to be replaced as indicated by the strong induction of *fldX*. Moreover, our results suggest that the expression of *fldX* is subject to saturation and reached its maximum expression level. This is evident when comparing the different oxidative stress conditions during cultivation without iron. In the background of oxidative stress, *fldX* is most likely expressed at maximum since the additional depletion of iron does not lead to a further increase in *fldX* transcription. The increased expression of *fldX* under both iron deficiency and various oxidative stress conditions and in combination is of particular interest because iron availability and oxidative stress are directly related. Oxidative stress can lead to the oxidation of $Fe^{2+}$ to $Fe^{3+}$ and thus significantly limit iron availability for *C. difficile*. However, our study shows that even at very high iron concentrations, oxidative stress leads to a significant increase in *fldX* transcription after a very short time. Thus, it is very likely that both iron limitation and oxidative stress are direct signals for *fldX* expression.

The results of the transcriptional analysis of *fldX* are confirmed by mass spectrometry data at protein level. Both the increase in FldX due to oxidative stress and the iron limitation-dependent increase are supported by proteomics data. Therefore, our study indicates that, indeed, the amount of FldX protein is increased in the cell. In addition to previous studies that demonstrated the increase of FldX under iron deficiency, our study now suggests a dual role of FldX, first, as a substitute for ferredoxins in iron deficiency (12) and second, as a possible detoxifying important electron transfer protein in the oxidative stress response.

For *CD2825*, the slot blot analyses clearly showed a significant increase in transcription under oxidative stress. $O_2$, $H_2O_2$, and $O_2^-$ influenced the expression of *CD2825* under normal and reduced iron concentration. The quantitative analysis via RT-qPCR confirmed the increased transcription rate under $H_2O_2$ stress during cultivation with 14.4 µM Fe and 0.2 nM Fe. However, no increase in *CD2825* transcription could be detected for $O_2$ and $O_2^-$ stress. In complete iron absence, induction of *CD2825* transcription fails, even under $H_2O_2$ stress. Possibly, the signal for increased *CD2825* expression is not $H_2O_2$ but the hydroxyl radical (44). This highly reactive radical is formed in the presence of ferrous iron by the Fenton reaction, which describes the reaction of $Fe^{2+}$ and $H_2O_2$ to form the toxic hydroxyl radical.

The transcriptional induction of *CD2825* could not be confirmed on protein level. Possibly, the stability of the *CD2825* mRNA is not very high, or due to post-transcriptional gene regulation mechanisms, it can no longer be translated. This hypothesis is supported by our half-life determination of the mRNAs of *fldX* and *CD2825*. It showed that the mRNA of *fldX* is significantly more stable than that of *CD2825*. This could explain why FldX could be detected at the protein level, but CD2825 could not. The mRNA stability and translation can be compromised by many post-transcriptional mechanisms. In particular, there are many gene-specific impairments of stability in the bacterial stress response (45). Nevertheless, our study determined a statistically significant increase in gene expression of *CD2825* during $H_2O_2$ stress. The physiological role and additional regulatory processes required for CD2825 protein expression need to be addressed in future studies. Another possible explanation for divergent transcript and protein data is the difference in waiting time before harvesting samples for transcriptional analyses and proteomics. While RNA samples were harvested immediately (10 min) after oxidative stress, the samples for proteome analyses were taken after 1 h to allow an adjustment of the proteome. It is possible that the increase in CD2825 after $H_2O_2$ stress is a very short-term effect and no longer detectable after 1 h.

## Conclusion and outlook

Our study is the first global investigation of flavodoxins in the human pathogen *C. difficile*. With seven putative flavodoxins, an extraordinarily high copy number of these proteins is annotated for *C. difficile*. We examined the expression of all putative flavodoxins under different growth and stress conditions and found most flavodoxins expressed in exponential growth phase, with *floX* showing particularly high transcription.

FldX was found to be significantly expressed on transcript- and protein level both under oxidative stress conditions and iron deficiency. An increased transcription of CD2825 could be determined especially under $H_2O_2$ stress, which was, however, not reflected on protein level.

Future studies should deal with the exact functional elucidation of the different flavodoxins. In particular, the search for interaction partners and the construction of knockout mutants could help to gain a better insight into their exact function. In addition, further growth studies could shed light on whether other stress conditions result in increased or reduced expression of selected flavodoxins.

Due to their essential importance as electron transfer proteins and their absence from eukaryotic, multicellular organisms, flavodoxins represent promising therapeutic targets to fight bacterial infections. In *Helicobacter pylori*, a highly specific flavodoxin inhibitor could already be found (45, 46). Such bacteria-specific flavodoxin inhibitors represent

good future treatment alternatives for broad-spectrum therapies, especially against intestinal pathogens such as *C. difficile* when sparing the natural intestinal microbiota is most favorable.

## MATERIALS AND METHODS

### Bioinformatic methods

A detailed *in silico* analysis of the eight putative flavodoxin sequences in *C. difficile* 630 (NCBI Taxonomy ID: 272563) was performed using the genome browser of the PRODORIC website (47). We focused on the exact localization in the genome, proximate genes, and composition of a possible operon structure calculated by the Operon-mapper web server (26). The multiple nucleotide and amino acid sequence alignments of the eight putative flavodoxins were generated using the default settings of Clustal Omega tool (48). A transcription terminator prognosis was generated with the "ARNold Finding Terminators" web service of the university Paris-Sud (27). An *in silico* analysis of the flavodoxin fold was carried out with the ColabFold platform (22). This open-source software uses the combination of homology search of MMseqs2 for multiple sequence alignment and AlphaFold2 for protein structure prediction. The domain and secondary structure analyses of the flavodoxin amino acid sequences were done applying the ScanProsite tool (23) and the web tool Porter 5 (21). The BLAST search tool (National Library of Medicine) was used to find homologs from other species and whether the eight flavodoxins are conserved across all *C. difficile* strains.

### Bacterial strains and growth conditions

All studies were carried out with *Clostridioides difficile* 630 (DSM No.: 27543) obtained from the German Collection of Microorganisms and Cell Cultures GmbH (DSMZ) (Braunschweig, Germany). *C. difficile* 630 was cultured in *C. difficile* minimal medium [according to reference (49)] under anaerobic conditions using an anaerobic chamber from Don Whitley Scientific Ltd. (Bingley, England). Three different iron concentrations (14.4 µM, 0.0002 µM, and no iron) were used, and $FeSO_4$ was supplemented, respectively. For no iron conditions, the chelator 2,2′-dipyridyl was added to the main culture at a final concentration of 75 µM. Precultures were inoculated with different volumes of germinated spores and were cultivated in CDMM with full iron concentration overnight. On the next morning, precultures with an optical density ($OD_{600}$) between 0.9 and 1.1 were used for inoculation of the main cultures. To prevent a distortion of the final iron concentration, precultures were centrifuged (8,500 rpm, 5 min), and pellets washed with CDMM without iron. Main cultures were inoculated to an $OD_{600}$ of 0.05. Cells were grown to exponential growth phase to an $OD_{600}$ of 0.4 to 0.5. Cultures without iron supplementation grew significantly slower but were further treated at the same time points as other cultures.

For RNA preparation, main cultures were split into five subcultures each and were treated with 5 % oxygen, 0.4 mM hydrogen peroxide, and 5 mM paraquat with and without additional 5 % $O_2$ treatment. One subculture remained untreated (control). After 10 min of treatment, 40 mL of the samples was cooled down with liquid nitrogen and centrifuged in gas-tight polypropylene tubes for 3 min at 8,500 rpm at 4°C. Cell pellets were stored at −80°C for later RNA preparation.

For protein samples, main cultures were split into five subcultures and were treated as samples for RNA preparation. After 10 min, the $H_2O_2$-treated subculture was centrifuged and resuspended in fresh CDMM with the appropriated iron concentration to avoid degradations due to permanent $H_2O_2$ treatment. All subcultures were cultivated for one additional hour under anaerobic conditions at 37°C and were centrifuged at 8,500 rpm for 5 min at 4°C, subsequently. Cell pellets were stored at −80°C for later protein extraction.

For flavodoxin RNA profiling along the growth curve, *C. difficile* 630 precultures and main culture were prepared as described in CDMM with an iron concentration of 14.4 µM. Samples were taken along the growth curve in exponential growth phase (at $OD_{600}$ = 0.4), at transient phase, early stationary phase, late stationary phase (12 h after inoculation), and death phase (24 h after inoculation). At each sampling point, 16 $OD_{600}$ units were harvested by filling the appropriated volume in gas tight polypropylene tubes. Samples were cooled down in liquid nitrogen, following centrifugation for 3 min at 8,500 rpm at 4°C. Cell pellets were stored at −80°C for further RNA preparation.

## cfu counting assays

*Clostridioides difficile* 630 cultures were grown as described before, and samples were taken at five different time points, including exponential, transient, early stationary, 12 h of growth, and 24 h of growth. At each time point, 100 µL of culture was plated on BHI agar in different dilutions ($10^{-1}$ to $10^{-5}$) according to the optical density, in two technical replicates. For the determination of cfu of germinated spores, 100 µL of each time point was incubated at 65°C for 20 min and, subsequently, plated on BHI agar with 0.01 % taurocholate in different dilutions according to the growth phase. All cfu plates were incubated at 37°C, and colonies counted after 24 h.

## Isolation of *C. difficile* nucleic acids

Using the "Bacterial Genomic DNA Isolation Kit" (NORGEN Biotek Corp., Ontario, Canada), chromosomal DNA was extracted from 1 mL of BHI medium *C. difficile* 630 culture. The manufacturer's protocol was followed, and the obtained chromosomal DNA was stored at −80°C. For cell lysis and RNA isolation, TRIzol reagent (Invitrogen, Thermo Fisher Scientific, Waltham, MA, USA) was used according to the manufacturer's protocol (50). RNA was solubilized in diethyl pyrocarbonate-treated water to a concentration of 1 µg/µL and was stored at −80°C. For quality control, an RNA gel was performed.

## Transcriptional profiling

For seven flavodoxins and the 5S rRNA gene, an individual RNA probe was constructed. Therefore, a PCR fragment of each gene was prepared using chromosomal DNA of *C. difficile* 630 as template with corresponding primers (listed in Table S6). Digoxigenin-labeled RNA probes were obtained, and Northern blot analyses carried out as previously described (51). For slot blot analyses, 5 µg RNA of each sample was blotted on a nylon membrane using a Bio-Dot microfiltration apparatus (Biorad Laboratories Ltd., Watford, England). The RNA was covalently linked to the membrane using UV light (120 $J/cm^2$). After washing, blocking and detection steps were carried out as previously described for Northern blot analyses (51). Signals were detected using a FUSION Solo 7S Imaging system and calculated with the Bio1D analysis software (Vilber Lourmat, Collégien, France).

Transcription of flavodoxins *CD1999* (*fldX)* and *CD2825* was quantified by RT-qPCR in three biological replicates with three technical replicates each. Raw data are provided in Tables S7 and S8. As a reference gene for normalization, *codY* was used due to its stable expression across the different iron stress conditions, oxidative stress conditions, and the combination of both stress conditions. cDNA was synthesized as previously described (52). The qPCR reactions were performed using the Luna Universal qPCR Master Mix (New England Biolabs GmbH, Frankfurt am Main, Germany) and primers given in Table S6 on a qTower[3] quantitative PCR thermocycler (Analytik Jena AG, Jena, Germany) (initial step at 50°C for 120 s; 95°C for 120 s; 41 cycles of 15 s 95°C, 60 s 60°C). The RT-qPCR data were analyzed using the qPCRsoft 4.1 (Analytik Jena AG, Jena, Germany).

## Half-life determination of RNA

For half-life determination of the RNA, *C. difficile* 630 cultures were grown to an OD of 0.4 in CDMM and then stressed with 0.4 mM $H_2O_2$ for 10 min. Subsequently, RNA synthesis was stopped by the addition of 200 µg/mL rifampicin. And, 20 mL samples were harvested after 0, 2, 4, 8, 16, and 32 min each by centrifugation at 8,500 rpm for 3 min. RNA extraction and Northern blot were performed as previously described. Northern blots are provided in Fig. S4. The signal intensities were measured and statistically evaluated as described for the slot blot evaluation. We used the signal intensities of the 5S RNA for normalization. For the calculation of half-lives, the log ratios of the normalized values (log2 FC) of the respective time points to the normalized value of time point 0 min were formed. These values were plotted against time, and the slope was calculated. The half-life was then calculated from the negative reciprocal of the slope (53). For the calculation of the slope, only those values were included that were approximately on the regression line. The half-life of *fldX* and *CD2825* was calculated based on three biological replicates.

## Protein extraction and MS sample preparation

For intracellular protein extraction, cell pellets were resuspended in 1 mL TE buffer (10 mM Tris, 1 mM EDTA, pH 8.0) followed by mechanical disruption with 400 µL glass beads (0.1 mm, Scientific Industries, Inc., Bohemia, NY, USA) in three homogenization cycles (each 30 s, 6.5 m/s) using a FastPrep-24 5G (MP Biomedicals, LLC, Irvine, CA, USA). Cell debris and glass beads were removed by centrifugation at $12,000 \times g$ for 15 min at 4°C. The resulting intracellular protein extracts were stored at −80°C.

Protein concentrations were determined using Roti Nanoquant (Carl Roth, Karlsruhe, Germany) according to the manufacturer's protocol. Also, 50 µg of proteins was prepared for mass spectrometry using S-trap spin columns (Protifi, Huntington, NY, USA) according to the manufacturer's protocol with the following modifications. For trypsin digestion, samples were incubated for 3 h at 47°C. For the three elution steps of the peptides, a 50-mM TEAB (Sigma, St. Louis, USA) solution was used as elution buffer 1, a 0.1% acetic acid solution as elution buffer 2, and a solution of 60% acetonitrile containing 0.1% acetic acid as elution buffer 3. Eluted peptides were desalted using 100 µL Pierce C18 Tips (Thermo Fisher Scientific, Waltham, MA, USA). Desalted peptides were vacuum dried and stored at −20°C.

## MS and PRM analyses

Flavodoxin expression on protein level during exponential growth phase was analyzed via LC-ESI-MS/MS. Therefore, we used protein samples of *C. difficile* 630 cultivated in CDMM with an iron concentration of 14.4 µM and harvested at an $OD_{600}$ of ~0.9.

All mass spectrometric experiments were performed on a Q-Exactive HF mass spectrometer coupled to an EASY nLC 1200 (Thermo Fisher Scientific, Waltham, MA, USA). The LC was equipped with an in-house built 20 cm × 100 µm reversed-phase (3 µm diameter C18 particles, ReproSil-Pur 120 C18-AQ, Dr. Maisch, Ammerbuch, Germany) column with an integrated emitter tip. Desalted peptides were dissolved in 12 µL solvent A (0.1% acetic acid in water), and about 50 µg of digested total protein amount per run was loaded onto the column. Elution was performed using a non-linear 100-min gradient of solvent B (0.1% acetic acid, 80% acetonitrile in water) in solvent A at a flow rate of 300 nL/min. The eluate from the column was injected into the mass spectrometer via the nano-ESI source, which was operated at 3.5 kV in positive mode. Survey scans in the Orbitrap were recorded at a resolution of 60,000 in the range of 333 to 16,500 m/z. The 15 most intense peaks per scan cycle were selected for fragmentation. Dynamic exclusion of precursor ions was enabled (30 s); single-charged ions as well as ions with unknown charge state were rejected. Internal lock mass calibration was enabled (lock mass 445.12003 Da).

LC-MS/MS data were searched against a strain-specific protein database [3,762 entries, obtained from Uniprot on 15 March 2021 (UP000001978)] using the

Andromeda-based search engine MaxQuant [(54), version 1.6.17.0]. Protein abundance was assessed by the Intensity-Based Absolute Quantification Index (iBAQ) (55). We used the formula $m_i = iBAQ_i \cdot M_i / \Sigma(iBAQ_j \cdot M_j)$ to calculate the relative mass of protein present ($m_i$) with the theoretical molecular weight $M_i$ of a protein $i$ (56).

The two putative flavodoxins FldX and CD2825 from *C. difficile* 630 were quantified from cells grown under varying iron concentrations and stress conditions by LC-ESI-MS/MS using PRM.

For the PRM experiment, each MS cycle consisted of a survey scan (333 to 1,650 Th mass range; 60,000 resolution at 200 m/z; 3e6 predictive automatic gain control target; max. 25 ms injection time; activated lock-mass correction) followed by up to 15 fragment ion scans (HCD at a normalized energy of 27; fixed first mass of 100 Th; 120,000 resolution at 200 m/z; 1e6 predicted automatic gain control target; max. 240 ms injection time; 1.4 m/z isolation window with 0.2 m/z offset) selected from a scheduled inclusion list (30 precursor entries, 12 min window size each).

Upon manual curation, PRM assays for two to five peptides per protein were established, and the obtained raw data were analyzed in Skyline [version 21.2 (57)]. For quantification, individual transition peak areas were summed peptide wise, and the resulting peptide intensities normalized against the survey scan's total ion counts (MS1 TIC). Furthermore, missing values were imputed assuming left-censored data, missing not at random by values from a value pool obtained by downshifting the remaining data with a narrow distribution. Peptides for which more than 80% of the data were missing were removed from all samples prior to summing peptide intensities to obtain protein abundances.

## Statistical analyses

Visualization of statistical analyses was performed using GraphPad Prism software (GraphPad Software Inc., La Jolla, CA).

For the determination of the half-life time of RNAs, we normalized the intensities with the signal of the 5S rRNA and calculated the fold change against the 0-min control sample. The half-life time was computed using the regression graph.

To determine the flavodoxin expression along the growth curve of *C. difficile* 630, fold changes between transient phase, stationary phase, 12-h growth, and 24-h growth were calculated against the expression of the corresponding flavodoxin in the exponential growth phase. Differences in transcription were tested for statistical significance using two-way ANOVA. α was set to 0.05. The RT-qPCR quantitative data analyses were based on the Pfaffl method (58). For slot blot analyses of transcription of flavodoxins under oxidative stress and iron limitation, as well as for RT-qPCR and PRM analyses of FldX and CD2825, expression fold changes were calculated, and statistical significance was checked using multiple *t*-testing with α of 0.05.

## ACKNOWLEDGMENTS

This work was supported by the German Research Foundation (231396381/GRK1947 to S.S. and 453440095 to S.S.) and by a Käthe-Kluth-scholarship of the University of Greifswald granted to S.S.

## AUTHOR AFFILIATIONS

[1]Department of Microbial Physiology and Molecular Biology, Institute of Microbiology, University of Greifswald, Greifswald, Germany
[2]Department of Microbial Proteomics, Institute of Microbiology, University of Greifswald, Greifswald, Germany

## AUTHOR ORCIDs

Daniel Troitzsch ⓘ http://orcid.org/0000-0002-4603-7525

Robert Knop http://orcid.org/0009-0007-7274-8252
Susanne Sievers http://orcid.org/0000-0002-5457-2552

## FUNDING

| Funder | Grant(s) | Author(s) |
| --- | --- | --- |
| Deutsche Forschungsgemeinschaft (DFG) | 231396381/GRK1947 | Daniel Troitzsch |
| Deutsche Forschungsgemeinschaft (DFG) | 453440095 | Robert Knop |
| Universität Greifswald (University of Greifswald) | Käthe-Kluth-Scholarship | Daniel Troitzsch |

## AUTHOR CONTRIBUTIONS

Daniel Troitzsch, Conceptualization, Funding acquisition, Investigation, Project administration, Resources, Supervision, Visualization, Writing – review and editing, Data curation, Formal analysis, Validation, Writing – original draft | Robert Knop, Conceptualization, Investigation, Visualization, Data curation, Formal analysis, Validation, Writing – original draft | Silvia Dittmann, Investigation, Visualization, Data curation, Formal analysis, Methodology, Validation, Writing – original draft | Jürgen Bartel, Investigation, Visualization, Formal analysis, Methodology, Software | Daniela Zühlke, Formal analysis, Methodology, Software | Timon Alexander Möller, Formal analysis, Methodology | Linda Trän, Investigation, Formal analysis, Methodology | Thaddäus Echelmeyer, Investigation, Visualization, Formal analysis | Susanne Sievers, Conceptualization, Funding acquisition, Investigation, Project administration, Resources, Supervision, Visualization, Writing – review and editing

## DATA AVAILABILITY

The mass spectrometry proteomics data have been deposited to the ProteomeXchange Consortium via the PRIDE partner repository with data set identifier PXD041951.

## ADDITIONAL FILES

The following material is available online.

Supplemental Material

**Fig. S1 to S4 (Spectrum01895-23-S0001.pdf).** Supplementary figures.
**Table S1 (Spectrum01895-23-S0002.xlsx).** Flavodoxin alignment.
**Table S2 (Spectrum01895-23-S0003.xlsx).** Flavodoxin homologs.
**Table S3 (Spectrum01895-23-S0004.xlsx).** Alignment of *C. difficile* strains.
**Table S4 (Spectrum01895-23-S0005.xlsx).** Flavodoxin operon structure.
**Table S5 (Spectrum01895-23-S0006.xlsx).** MaxQuant result.
**Table S6 (Spectrum01895-23-S0007.xlsx).** Oligonucleotides.
**Table S7 (Spectrum01895-23-S0008.xlsx).** RT-qPCR of *fldX*.
**Table S8 (Spectrum01895-23-S0009.xlsx).** RT-qPCR of *CD2825*.

Open Peer Review

**PEER REVIEW HISTORY (review-history.pdf).** An accounting of the reviewer comments and feedback.

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
