## [Reviewer comments · Microbiology Spectrum]

Microbiology Spectrum

Characterizing the Flavodoxin Landscape in *Clostridioides difficile*

Daniel Troitzsch, Robert Knop, Silvia Dittmann, Jürgen Bartel, Daniela Zühlke, Timon Möller, Linda Trän, Thaddäus Echelmeyer, and Susanne Sievers

Corresponding Author(s): Susanne Sievers, Universität Greifswald

Review Timeline:

Submission Date:	May 5, 2023
Editorial Decision:	July 14, 2023
Revision Received:	October 27, 2023
Editorial Decision:	November 28, 2023
Revision Received:	December 21, 2023
Accepted:	December 23, 2023

Editor: Meera Unnikrishnan

Reviewer(s): The reviewers have opted to remain anonymous.

Transaction Report:

DOI: <https://doi.org/10.1128/spectrum.01895-23>

July 14, 2023

Dr. Susanne Sievers
Universität Greifswald
Institute of Microbiology
Felix-Hausdorff-Str. 8
Greifswald
Germany

Re: Spectrum01895-23 (Characterizing the Flavodoxin Landscape in *Clostridioides difficile*)

Dear Dr. Susanne Sievers:

Thank you for submitting your manuscript to Microbiology Spectrum.

We have received comments back from two expert reviewers which are included below. Although they find the paper interesting, they have requested additional experiments in support of some of the conclusions made. Concerns regarding the statistical analysis also need to be addressed. Please submit a revised manuscript addressing the Reviewers' comments.

Link Not Available

Sincerely,

Meera Unnikrishnan

Journals Department
Reviewer comments:

Reviewer #2 (Comments for the Author):

In the manuscript titled "Characterizing the flavodoxin landscape in *Clostridioides difficile*" by Troitzsch et al., the authors identified and characterized seven flavodoxin-like proteins. By performing in silico whole genome analysis, the authors identified eight flavodoxin-like proteins in *C. difficile*. Troitzsch et al. compared these proteins to known flavodoxins in sequence

conservation, predicted tertiary structure, and gene synteny. They went on and used Northern blot to validate and screen the expression of the outlined proteins under various growth conditions. Lastly, Troitzsch et al., quantified the transcriptional and translational levels of the flavodoxin-like proteins under distinct stressed conditions and found that FldX was induced by iron limitation and oxidative stress. While it is interesting to investigate the flavodoxin-landscape in *C. difficile*, from microbial physiology and pathogenesis standpoints, the lack of any in-depth characterization or mechanistic insights into the roles of these proteins in *C. difficile* biology dampens the enthusiasm for the paper.

Major points:

1. The authors identified eight flavodoxin-like proteins, but the evidence supporting that they are actually flavodoxins is not very concrete, especially given the lack of sequence conservation amongst these eight proteins. Further biochemical or genetic evidence will be needed to support the authors' claim. While the genetics of *C. difficile* is understandably challenging, assays using purified recombinant proteins would greatly strengthen the authors' conclusion.
2. It is very interesting that fldX is coregulated by iron availability and oxidative stress. However, it is unclear whether the observed phenotype is solely due to the presence of the oxidants, iron being oxidized by the added oxidants and the iron availability/solubility plummets, or it is because of the Fenton reaction that amplifies the phenotypes. The latter is consistent with the observation that adding DP abolished the induction caused by O₂ exposure. While the differential induction observed when different oxidants were added suggests that there is some specificity to the phenotypes, further explanation is needed to clarify the potential underlying mechanisms.

Minor:

1. *C. difficile* is not known to produce siderophores, how does it grow when "0 nM" of iron is present in the media?

Reviewer #3 (Comments for the Author):

This manuscript by Troitzsch et al. tends to characterize the numerous flavodoxins found in *C. difficile*. The authors looked for flavodoxins in the genome of reference strain 630, their genetic organization, and tested their expression level both under oxidative stresses and iron limitation. Additional proteomic data attests the importance of one of the described flavodoxins. The flow of experiments is logical and conclusions are convincing. Some comments are listed below:

Major comments

- Authors must integrate TSS data from Soutourina et al. 2020 in their analysis doi: 10.3389/fmicb.2020.01939.
- Authors should comment the expression of the flavodoxin genes in published transcriptomic analysis especially fldX in Giordano et al. 2018 doi: 10.1093/femspd/fty010
- Fig. 5: Result at 12h, though analyzed in the discussion is intriguing. Is an experimental issue possible? Has another housekeeping-gene been blotted as control of quality/quantity of RNA? Or RT-PCR performed on these specific sample to confirm the results? Authors could add information of total CFUs and spores to S1 to support their hypothesis.
- 3 biological replicates are not enough for RT-qPCR analysis. Statistical analysis based on the rank must be performed on at least 4 replicates.

Minor comments

- In the abstract, not clear how flavodoxins could represent a therapeutic target (even though explained in the discussion)
- What is the reference for the many cases of lethal outcomes line 27? Sentence should be moderated
- P6 Line 11 and in material and methods: add the accession number of the used strain
- To facilitate reading, reorganize Fig. 2B according to the text with first line short model, second line short model with additional helices, third line long model and long CD flavodoxins and 4th line the two differing flavodoxins.
- Fig 2B, show the 4Fe-4S centers of CD3121
- P7 and Fig 3. Is the genomic organization conserved outside *C. difficile* strain?
- Fig. 3: indicate the names used in the text in the figure and not only in the legend. Supplementary data should present the annotation of presented CD genes and published TSS of these regions.
- P8 Line 6: remove last sentence. This is questioning. Has the specificity of the RNA probe binding been tested? How many times gels from Fig 4 have been performed? Methylene blue staining is not the right control and covers the signal of some RNAs of interest. Membrane should be hybridized with 5S RNA probes for instance and quantification should be performed on at least 3 images relative to 3 biological replicates. Signal of the RNAs of interest should be reported to the signal of the control probe.
- Half-life of RNAs would be a real plus especially for CD1999 and its different forms and for CD2825 that is discussed P20 line 11.
- Fig. 6B is hard to read. Authors could choose to present only the two control conditions i.e. the two first lines and keep the rest as supplementary data. floX is not discussed despite the relatively high FC at 0.2 nM Fe probably because of the statistical analysis but in that case why discuss CD2207? Remove however line 6 p11.
- Fig. 9A FC numbers or a zoom with a change of scale would be necessary in the no iron condition to visualize the importance of the stars
- Fig. 9B and 10B the legend is not the same as Fig. 7B and 8B whereas it is in A panels.
- In discussion p14, are lines 16-17 p14 related to ref 6? If not add reference at the end of the sentence if yes put ref (6) at the end of the paragraph.
- In discussion p14, what if the reference of sentence line 28-29? If this is this study add data as supplementary information.

Authors should give the names of a few species concerned by the high number of flavodoxins.

- In discussion, strange that paragraph titles appear only after the 4th paragraph
- In discussion, line 4, has-it-been demonstrated that residual amounts of iron are sufficient for the formation of 2Fe-2S clusters and ferredoxins? How? Where?
- In material and methods and S1: change A600 OD600
- In material and methods, steps are missing in the isolation of nucleic acids part e.g. what DNase treatment has been applied? How DNA contamination has been assessed? Legend NTC in S7 and S8.
- In statistical analysis and Fig. 7-8-9-10 precise what the line stands for e.g. median.

Edition comments

Comas are missing at different places.

Abstract/Importance

- Line 3: remove even
- Line 29: remove With

Introduction

- P4 Line 12: remove already
- P4 Line 23: remove and
- P4 Line 23: remove comprising mostly only
- P4 Line 27: remove whereas ; change have having
- P5 Line 3: change are have been
- P5 Line 17: remove the s from bioinformatics analysis
- P5 Line 18: add the in front of eight flavodoxins

Results

- P6 Line 4: remove for
- P6 Line 12: remove approximately
- P6 Line 20 in the parenthesis, organize data with semi-column between the two types of flavodoxins and the addition of & for instance
- P6 Line 27: inverse features and FloX
- P6 Line 28: add s to beta-strand
- P7 Line 7: write BLAST with capital letters
- P7 Line 13: remove is
- P7 Line 14: change coding for encodes
- P7 Line 15: add s to oxidoreductase
- P7 Line 17: change more identical with closer to
- P7 Line 18: remove taken together ; remove the protein
- P8 Line 9: Fig 3 Fig 4
- P8 Line 10: remove interestingly ; change just only
- P8 Line 29: add reference to Fig. 4
- P9 Line 4: remove but not for the larger one : change at under
- P9 Line 9: change decided to screen the flavodoxin we screened flavodoxin
- P10 Line 5: remove most likely
- P10 Line 29: remove capital letter of fldX
- P12 Line 4: remove capital letter of fldX

Discussion

- P15 Line 11: add FldX gene number
- P15 Line 22: remove s from needs
- P15 Line 25: space missing after 2021
- P16 Line 2: wrong place for the verb
- P16 Line 4: idem
- P18 Line 12: add be after cannot ; change but and
- P18 Line 15-16: check tenses and merge the two sentences
- P18 Line 19-20: check
- P18 Line 26: stress stresses
- P19 Line 7: remove Now e.g. In this condition
- P19 Line 12: remove already
- P19 Line 15: remove thus
- P20 Line 13: remove mostly

Material and Methods

- P22 Line 1: BLAST capital letters
- P24 Line 10: RT-PCR, q missing

Staff Comments:

Preparing Revision Guidelines

Please return the manuscript within 60 days; if you cannot complete the modification within this time period, please contact me. If you do not wish to modify the manuscript and prefer to submit it to another journal, please notify me of your decision immediately so that the manuscript may be formally withdrawn from consideration by Microbiology Spectrum.

July 2023 - Review
Characterizing the Flavodoxin landscape in *Clostridioides difficile*

This manuscript by Troitzsch *et al.* tends to characterize the numerous flavodoxins found in *C. difficile*. The authors looked for flavodoxins in the genome of reference strain 630, their genetic organization, and tested their expression level both under oxidative stresses and iron limitation. Additional proteomic data attests the importance of one of the described flavodoxins.

The flow of experiments is logical and conclusions are convincing. Some comments are listed below:

Major comments

- Authors must integrate TSS data from Soutourina *et al.* 2020 in their analysis doi: 10.3389/fmicb.2020.01939.
- Authors should comment the expression of the flavodoxin genes in published transcriptomic analysis especially *fldX* in Giordano *et al.* 2018 doi: 10.1093/femspd/fty010
- Fig. 5: Result at 12h, though analyzed in the discussion is intriguing. Is an experimental issue possible? Has another housekeeping-gene been blotted as control of quality/quantity of RNA? Or RT-PCR performed on these specific sample to confirm the results? Authors could add information of total CFUs and spores to S1 to support their hypothesis.
- 3 biological replicates are not enough for RT-qPCR analysis. Statistical analysis based on the rank must be performed on at least 4 replicates.

Minor comments

- In the abstract, not clear how flavodoxins could represent a therapeutical target (even though explained in the discussion)
- What is the reference for the many cases of lethal outcomes line 27? Sentence should be moderated
- P6 Line 11 and in material and methods: add the accession number of the used strain
- To facilitate reading, reorganize Fig. 2B according to the text with first line short model, second line short model with additional helices, third line long model and long CD flavodoxins and 4th line the two differing flavodoxins.
- Fig 2B, show the 4Fe-4S centers of CD3121
- P7 and Fig 3. Is the genomic organization conserved outside *C. difficile* strain?
- Fig. 3: indicate the names used in the text in the figure and not only in the legend. Supplementary data should present the annotation of presented CD genes and published TSS of these regions.
- P8 Line 6: remove last sentence. This is questioning. Has the specificity of the RNA probe binding been tested? How many times gels from Fig 4 have been performed? Methylene blue staining is not the right control and covers the signal of some RNAs of interest. Membrane should be hybridized with 5S RNA probes for instance and quantification should be performed on at least 3 images relative to 3 biological replicates. Signal of the RNAs of interest should be reported to the signal of the control probe.
- Half-life of RNAs would be a real plus especially for *CD1999* and its different forms and for *CD2825* that is discussed P20 line 11.

- Fig. 6B is hard to read. Authors could choose to present only the two control conditions *i.e.* the two first lines and keep the rest as supplementary data. *floX* is not discussed despite the relatively high FC at 0.2 nM Fe probably because of the statistical analysis but in that case why discuss *CD2207*? Remove however line 6 p11.
- Fig. 9A FC numbers or a zoom with a change of scale would be necessary in the no iron condition to visualize the importance of the stars
- Fig. 9B and 10B the legend is not the same as Fig. 7B and 8B whereas it is in A panels.
- In discussion p14, are lines 16-17 p14 related to ref 6? If not add reference at the end of the sentence if yes put ref (6) at the end of the paragraph.
- In discussion p14, what if the reference of sentence line 28-29? If this is this study add data as supplementary information. Authors should give the names of a few species concerned by the high number of flavodoxins.
- In discussion, strange that paragraph titles appear only after the 4th paragraph
- In discussion, line 4, has-it-been demonstrated that residual amounts of iron are sufficient for the formation of 2Fe-2S clusters and ferredoxins? How? Where?
- In material and methods and S1: change A₆₀₀ → OD₆₀₀
- In material and methods, steps are missing in the isolation of nucleic acids part *e.g.* what DNase treatment has been applied? How DNA contamination has been assessed? Legend NTC in S7 and S8.
- In statistical analysis and Fig. 7-8-9-10 precise what the line stands for *e.g.* median.

Edition comments

Comas are missing at different places.

Abstract/Importance

- Line 3: remove even
- Line 29: remove With

Introduction

- P4 Line 12: remove already
- P4 Line 23: remove and
- P4 Line 23: remove comprising mostly only
- P4 Line 27: remove whereas ; change have → having
- P5 Line 3: change are → have been
- P5 Line 17: remove the s from bioinformatics analysis
- P5 Line 18: add the in front of eight flavodoxins

Results

- P6 Line 4: remove for
- P6 Line 12: remove approximately
- P6 Line 20 in the parenthesis, organize data with semi-column between the two types of flavodoxins and the addition of & for instance
- P6 Line 27: inverse features and FloX
- P6 Line 28: add s to beta-strand
- P7 Line 7: write BLAST with capital letters
- P7 Line 13: remove is
- P7 Line 14: change coding for → encodes

- P7 Line 15: add s to oxidoreductase
- P7 Line 17: change more identical with → closer to
- P7 Line 18: remove taken together ; remove the protein
- P8 Line 9: Fig 3 → Fig 4
- P8 Line 10: remove interestingly ; change just → only
- P8 Line 29: add reference to Fig. 4
- P9 Line 4: remove but not for the larger one : change at → under
- P9 Line 9: change decided to screen the flavodoxin → we screened flavodoxin
- P10 Line 5: remove most likely
- P10 Line 29: remove capital letter of *fldX*
- P12 Line 4: remove capital letter of *fldX*

Discussion

- P15 Line 11: add FldX gene number
- P15 Line 22: remove s from needs
- P15 Line 25: space missing after 2021
- P16 Line 2: wrong place for the verb
- P16 Line 4: idem
- P18 Line 12: add be after cannot ; change but → and
- P18 Line 15-16: check tenses and merge the two sentences
- P18 Line 19-20: check
- P18 Line 26: stress → stresses
- P19 Line 7: remove Now → e.g. In this condition
- P19 Line 12: remove already
- P19 Line 15: remove thus
- P20 Line 13: remove mostly

Material and Methods

- P22 Line 1: BLAST capital letters
- P24 Line 10: RT-PCR, q missing

Re: Spectrum01895-23 (Characterizing the Flavodoxin Landscape in *Clostridioides difficile*)

Dear Dr. Susanne Sievers:

Thank you for submitting your manuscript to Microbiology Spectrum.

We have received comments back from two expert reviewers which are included below. Although they find the paper interesting, they have requested additional experiments in support of some of the conclusions made. Concerns regarding the statistical analysis also need to be addressed. Please submit a revised manuscript addressing the Reviewers' comments.

<https://spectrum.msubmit.net/cgi-bin/main.plex?el=A7QF6CJau7A7ELXS4F2A9ftdmxcT49ZrOaLeMmrqeseaFgZ>

Sincerely,

Meera Unnikrishnan

Journals Department
Reviewer comments:

Reviewer #2 (Comments for the Author):

In the manuscript titled "Characterizing the flavodoxin landscape in *Clostridioides difficile*" by Troitzsch et al., the authors identified and characterized seven flavodoxin-like proteins. By performing in silico whole genome analysis, the authors identified eight flavodoxin-like proteins in *C. difficile*. Troitzsch et al. compared these proteins to known flavodoxins in sequence conservation, predicted tertiary structure, and gene synteny. They went on and used Northern blot to validate and screen the expression of the outlined proteins under various growth conditions. Lastly, Troitzsch et al., quantified the transcriptional and translational levels of the flavodoxin-like proteins under distinct stressed conditions and found that FldX was induced by iron limitation and oxidative stress. While it is interesting to investigate the flavodoxin-landscape in *C. difficile*, from microbial physiology and pathogenesis standpoints, the lack of any in-depth characterization or mechanistic insights into the roles of these proteins in *C. difficile* biology dampens the enthusiasm for the paper.

*We appreciate that the reviewer considers flavodoxins in *C. difficile* to be an interesting subject. Of course we agree, that all putative flavodoxins of *C. difficile* should be characterized and their specific role in physiology and virulence should be elucidated in the future. However, in our opinion this goes beyond the scope of a single paper. This work should be a first overview study and the first ever to deal more intensively with flavodoxins and the unusually high number of these electron-transferring small proteins in the pathogen *C. difficile*. Although mechanistic data are not yet available, valuable data on the expression of individual flavodoxins were collected and virulence-relevant conditions were identified leading to massive induction of individual flavodoxins. In summary: (1) a complete picture of all flavodoxins in *C. difficile* has been established and (2) flavodoxins that are of particular interest from an infectiological point of view have been identified.*

Major points:

1. The authors identified eight flavodoxin-like proteins, but the evidence supporting that they are actually flavodoxins is not very concrete, especially given the lack of sequence conservation amongst these eight proteins. Further biochemical or genetic evidence will be needed to support the authors' claim. While the genetics of *C. difficile* is understandably

challenging, assays using purified recombinant proteins would greatly strengthen the authors' conclusion.

There is no 100% certainty that the 7 putative flavodoxins in this work are really flavodoxins until each one and its interaction partners have been characterized in detail and structurally elucidated. This was not the intention of this work and would be far beyond the scope of a single paper. The fact that flavodoxins show little sequence similarity is not an indication that they are not flavodoxins. Flavodoxins do not have a specific primary sequence but a characteristic fold (secondary and tertiary structure) that was predicted for all proteins. Overexpression of the proteins would be possible, but is not sufficient to prove that they are flavodoxins, because to prove their electron transfer ability, the interaction partners would also have to be known, which will be addressed in our future studies.

Nevertheless, we agree with the reviewer that we need to make clear in the manuscript that we discuss only putative flavodoxins. Therefore, even more care was taken to refer to the flavodoxins always as putative flavodoxins.

2. It is very interesting that fldX is coregulated by iron availability and oxidative stress. However, it is unclear whether the observed phenotype is solely due to the presence of the oxidants, iron being oxidized by the added oxidants and the iron availability/solubility plummets, or it is because of the Fenton reaction that amplifies the phenotypes. The latter is consistent with the observation that adding DP abolished the induction caused by O₂ exposure. While the differential induction observed when different oxidants were added suggests that there is some specificity to the phenotypes, further explanation is needed to clarify the potential underlying mechanisms.

The reviewer is correct that further studies, for instance biochemical and structural analyses on purified flavodoxins, are needed to determine the exact mechanism of induction of flavodoxins. Still, our data strongly indicated different flavodoxins to have different stimuli. In the case of fldX, we believe that it is induced by both iron limitation and oxygen, since induction by oxygen also works in a medium with excess Fe²⁺ and can be detected after only 10 min. The Fenton reaction would be even stronger in the presence of H₂O₂, but fldX is more induced by oxygen than in the presence of H₂O₂. It would be conceivable that the flavodoxin FldX replaces a certain ferredoxin (e.g. that of the FNR complex), which can no longer be produced during iron deficiency and is very sensitive to oxidation and thus no longer produced even if sufficient iron is present. Our results indicate that for CD2825 the oxidative stress stimulus is H₂O₂ rather than O₂. These observed differences and conclusions are discussed in the manuscript.

Minor:

1. *C. difficile* is not known to produce siderophores, how does it grow when "0 nM" of iron is present in the media?

We don't know how C. difficile manages to grow in 0 nM iron, but obviously addition of an iron chelator is necessary to induce iron limitation as was shown by other groups before (Ho et al. 2015, Cernat et al. 2012, Hastie et al. 2018).

We have analyzed the growth of C. difficile in different iron concentrations (Fig. S1) and on that base decided to carry out our experiments with the concentrations given in the paper.

Reviewer #3 (Comments for the Author):

This manuscript by Troitzsch et al. tends to characterize the numerous flavodoxins found in *C. difficile*. The authors looked for flavodoxins in the genome of reference strain 630, their genetic organization, and tested their expression level both under oxidative stresses and iron limitation. Additional proteomic data attests the importance of one of the described flavodoxins.

The flow of experiments is logical and conclusions are convincing. Some comments are listed below:

Major comments:

- Authors must integrate TSS data from Soutourina et al. 2020 in their analysis doi: 10.3389/fmicb.2020.01939.

We appreciate the reviewers indication of the work of Soutourina et al. published in 2020. We have now included the TSS data of Soutourina et al. and of Fuchs et al. (2021) and integrated both datasets into our analysis resulting in an expansion of information given in supplemental table S4.

- Authors should comment the expression of the flavodoxin genes in published transcriptomic analysis especially *fldX* in Giordano et al. 2018 doi: 10.1093/femspd/fty010

*We thank the reviewer for pointing out the published transcriptome data of Giordano et al. 2018. In this publication, microarray analyses were performed on C. difficile 630 in BHI under control and under 2 % oxygen condition. The consideration of relative expressions showed that none of the flavodoxins studied is among the 25 most induced genes at 2 % oxygen condition. Flavodoxin expression data are given in the supplementary part. There, results for *fldX* in the three performed replicates are very different and do not match our observations made in CDMM at 5 % oxygen (table 1). We cited the work of Giordano et al. but could not compare their data with our data.*

Table 1: Microarray data analysis of *C. difficile* 630 by Giordano *et al.* 2018

	anaerobic			aerobic (2 % oxygen)			FC			FC average
flavodoxin	replicate 1	replicate 2	replicate 3	replicate 1	replicate 2	replicate 3	replicate 1	replicate 2	replicate 3	
CD0810 (floX)	3,13674 5453	2,31619 8278	2,190917 11	2,49357 5923	2,34889 8634	2,70477 6047	0,79495 6416	1,01411 8116	1,23454 0565	1,014538 366
CD1458 (wrbA)	1,27186 1407	1,37145 2869	1,427712 319	0,38596 2107	0,32338 1278	0,33786 1133	0,30346 2394	0,23579 4671	0,23664 5106	0,258634 057
CD1679	- 1,94537 8714	- 1,78260 0401	- 2,244817 724	- 1,43348 6766	- 1,83768 5234	- 2,30222 075	0,73686 7714	1,03090 1392	1,02557 1353	0,931113 486
CD1999 (fldX)	2,27089 8048	2,32340 0367	- 0,663471 44	- 1,49344 0451	0,20615 5568	2,02969 0464	- 0,65764 3108	0,08873 0109	- 3,05919 7942	- 1,209370 314
CD2207	0,81505 7852	0,89710 5578	0,843452 663	0,68066 1249	0,65814 812	0,44723 1847	0,83510 7897	0,73363 5077	0,53023 9415	0,699660 797
CD2684	- 0,24842 3824	- 0,77338 1572	- 0,686146 796	0,49837 8265	0,22195 9259	0,35060 9581	- 2,00616 1313	- 0,28699 8381	- 0,51098 3339	- 0,934714 344
CD2825	0,98156 5899	1,28358 1531	1,550224 445	0,59193 6557	0,77701 2517	0,38249 7082	0,60305 3302	0,60534 7224	0,24673 6583	0,485045 703

- Fig. 5: Result at 12h, though analyzed in the discussion is intriguing. Is an experimental issue possible? Has another housekeeping-gene been blotted as control of quality/quantity of RNA? Or RT-PCR performed on these specific sample to confirm the results? Authors could add information of total CFUs and spores to S1 to support their hypothesis.

We thank the reviewer for the suggestion to complement our results with CFU and spore count data. We repeated the entire experiment and carried out CFU and spore counts in parallel in five biological replicates and included the obtained data in the manuscript. There is no housekeeping gene showing constant expression along the entire time course of this experiment. However, in this repeated experiment we have also blotted 5S rRNA. Its expression is constant till the 12 h time point, so we can exclude technical issues there. At the 24 h time point 5S rRNA levels decrease which is plausible in the death phase.

- 3 biological replicates are not enough for RT-qPCR analysis. Statistical analysis based on the rank must be performed on at least 4 replicates.

We agree with the reviewer that a higher number of replicates would increase the power of the statistical analysis. The number of biological replicates in this manuscript is based on the original publication of Pfaffl's method for the analysis of RT-qPCR results as described in the Materials and Methods section (Pfaffl MW. 2001. a new mathematical model for relative quantification in real-time RT-PCR. Nucleic Acids Res 29. doi:10.1093/nar/29.9.e45.). Other subsequent papers also used 3 biological replicates when using the Pfaffl method. Therefore, we also performed three replicates and reach statistical significance with our data which have been normalized by data of stable RNA of the housekeeping gene codY.

Minor comments:

- In the abstract, not clear how flavodoxins could represent a therapeutical target (even though explained in the discussion)

Unfortunately, the word limit of the abstract did not allow us to go into more detail about the therapeutic application. Therefore, we redrafted the sentence in the abstract and explain the statement in more detail in the discussion section.

- What is the reference for the many cases of lethal outcomes line 27? Sentence should be moderated

We replaced lethal outcome with severe outcome and refer to the most recent data of the infectious epidemiological yearbook of notifiable diseases in Germany in 2020. Although the incidence has been decreasing in recent years, CDI with a severe course still belongs to the top 20 notifiable diseases with a value of 1.9/100.000 inhabitants, even ahead of MRSA infections.

- P6 Line 11 and in material and methods: add the accession number of the used strain

We added the NCBI Taxonomy ID 272563 and the DSM Number 27543 of the used Clostridioides difficile 630 strain at the corresponding place in the results and methods part.

- To facilitate reading, reorganize Fig. 2B according to the text with first line short model, second line short model with additional helices, third line long model and long CD flavodoxins and 4th line the two differing flavodoxins.

We reorganized the picture composition of figure 2B, according to the reviewers recommendation and hope the figure is clearer now.

- Fig 2B, show the 4Fe-4S centers of CD3121

We revised the Figure 2B and added the localization of the two 4Fe-4S centers.

- P7 and Fig 3. Is the genomic organization conserved outside C. difficile strain?

We checked the genetic arrangement of the putative flavodoxin homologues of the best hits of Table S2 to see whether they are also conserved outside C. difficile. We found that, with a few

exceptions, there is no similar genetic neighborhood for the putative flavodoxin homologues in the organisms we looked at. We thank the reviewer for the hint, but decided against adding another supplement table with the genetic neighborhood of the putative flavodoxin homologues from other organisms to this paper, as it is not similar to the organization in C. difficile.

- Fig. 3: indicate the names used in the text in the figure and not only in the legend. Supplementary data should present the annotation of presented CD genes and published TSS of these regions.

We have changed the flavodoxin names in the figure and added the annotated TSS from Soutourina et al. and Fuchs et al., listed in the revised and expanded supplement table S4.

- P8 Line 6: remove last sentence. This is questioning. Has the specificity of the RNA probe binding been tested? How many times gels from Fig 4 have been performed? Methylene blue staining is not the right control and covers the signal of some RNAs of interest. Membrane should be hybridized with 5S RNA probes for instance and quantification should be performed on at least 3 images relative to 3 biological replicates. Signal of the RNAs of interest should be reported to the signal of the control probe.

We changed the last sentence and complemented the information by the comparison to the TSS annotated by Fuchs et al.. We checked the specificity of any of the approximately 300 nt long RNA probes by sequence alignment with the whole genome and could not find any similar region. For 100% certainty, we would have to do Northern blots with the deletion mutants which we do not have on hand yet. The methylene blue staining in figure 4 was only used to detect the gel loading and the structural integrity of the RNA. We agree with the reviewer that the methylene blue staining is not suited for quantification, but at this point we did not aim for a quantification of RNA. Figure 4 is mainly concerned with the experimental determination of RNA sizes and hence of a possible operon structure. Because we did not infer any quantitative statements from results shown in figure 4, there is no point of taking along another reference RNA or several replicates.

- Half-life of RNAs would be a real plus especially for CD1999 and its different forms and for CD2825 that is discussed P20 line 11.

We thank the reviewer for this valuable suggestion and carried out half-life time experiments for CD1999 (fldX) and CD2825. The data obtained resulted in a new figure (Fig 11) and were incorporated and discussed in the revised main manuscript.

- Fig. 6B is hard to read. Authors could choose to present only the two control conditions i.e. the two first lines and keep the rest as supplementary data. floX is not discussed despite the relatively high FC at 0.2 nM Fe probably because of the statistical analysis but in that case why discuss CD2207? Remove however line 6 p11.

We agree with the reviewer that Fig. 6 is a very complex figure. Fig. 6 is intended to show both the changes in flavodoxin expression under different oxidative stress conditions and the influence of different iron concentrations. Therefore, despite the complexity of the figure, we decided to leave the entire table in the manuscript. In our opinion, moving part of the figure to the Supplemental Material would not lead to an improved understanding of Fig. 6 but rather to a missing of important information.

Regarding the discussion of the individual flavodoxins, we wanted to focus primarily on those flavodoxins that showed a statistically relevant induction. We agree with the reviewer that in this case the discussion of CD2207 is confusing and have therefore removed the relevant sentence from the manuscript.

- Fig. 9A FC numbers or a zoom with a change of scale would be necessary in the no iron condition to visualize the importance of the stars

We added in Fig. 9A a zoom in the graph under no iron condition which increases the information content of this figure. We appreciate the reviewers suggestion.

- Fig. 9B and 10B the legend is not the same as Fig. 7B and 8B whereas it is in A panels.

The legends for Fig. 9B and 10B were changed to reflect legends of Fig. 7B and 8B now.

- In discussion p14, are lines 16-17 p14 related to ref 6? If not add reference at the end of the sentence if yes put ref (6) at the end of the paragraph.

Both lines belong to Ref. 6. The reference has been moved to the end of the paragraph.

- In discussion p14, what if the reference of sentence line 28-29? If this is this study add data as supplementary information. Authors should give the names of a few species concerned by the high number of flavodoxins.

Mentioned were results of the study by Campbell et al. A reference has been added at the end of the sentence. Furthermore, some example of microorganisms with 6 or more flavodoxins have been added.

- In discussion, strange that paragraph titles appear only after the 4th paragraph

The paragraph titles are intended to organize the discussion into logical sections. In particular, our own results should be structured here. We agree with the reviewer that it was strange that the discussion of our in-silico analyses did not have a paragraph title. This has been changed in the revised manuscript.

- In discussion, line 4, has-it-been demonstrated that residual amounts of iron are sufficient for the formation of 2Fe-2S clusters and ferredoxins? How? Where?

As far as we know, this could not be proven in any study so far. Nevertheless, our results show that although fldX is induced under low iron concentrations, it can be induced even more strongly in CDMM without iron. It has already been shown in other studies that flavodoxins often substitute for ferredoxins. Since fldX showed particularly strong induction in the presence of iron deficiency, we concluded that the low iron level of 0.2 nM appears to be sufficient for C. difficile to still form ferredoxins to a small extent. However, ferredoxins are already substituted to some extent by flavodoxin FldX. Under iron exclusion, [2Fe-2S] clusters are then no longer formed, which is why the amount of FldX increases again. To answer the reviewer's question: No, it was not shown that traces of iron are enough for ferredoxin formation, but in our discussion we discuss this as an explanation of the experimental data we obtained for fldX expression.

- In material and methods and S1: change A600 to OD600

Both in the manuscript and in figures, A600 was changed to OD600

- In material and methods, steps are missing in the isolation of nucleic acids part e.g. what DNase treatment has been applied? How DNA contamination has been assessed? Legend NTC in S7 and S8.

In RNA isolation with trizol, the RNA phase is cleanly separated from DNA and other cell components by using a phase separation process. To remove any residual DNA, we use a DNase treatment only immediately before transcribing the RNA into cDNA.

An explanation for NTC (no template control) was added in the legends.

- In statistical analysis and Fig. 7-8-9-10 precise what the line stands for e.g. median.

The line in the graphs shows the mean of each condition. An explanation was added to the figure captions.

Edition comments

Comas are missing at different places.

Abstract/Importance

- Line 3: remove even
- Line 29: remove With

Introduction

- P4 Line 12: remove already
- P4 Line 23: remove and

- P4 Line 23: remove comprising mostly only
- P4 Line 27: remove whereas ; change have to having
- P5 Line 3: change are to have been
- P5 Line 17: remove the s from bioinformatics analysis
- P5 Line 18: add the in front of eight flavodoxins

Results

- P6 Line 4: remove for
- P6 Line 12: remove approximately
- P6 Line 20 in the parenthesis, organize data with semi-column between the two types of flavodoxins and the addition of & for instance
- P6 Line 27: inverse features and FloX
- P6 Line 28: add s to beta-strand
- P7 Line 7: write BLAST with capital letters
- P7 Line 13: remove is
- P7 Line 14: change coding for to encodes
- P7 Line 15: add s to oxidoreductase
- P7 Line 17: change more identical with to closer to
- P7 Line 18: remove taken together ; remove the protein
- P8 Line 9: Fig 3 to Fig 4
- P8 Line 10: remove interestingly ; change just to only
- P8 Line 29: add reference to Fig. 4
- P9 Line 4: remove but not for the larger one : change at to under
- P9 Line 9: change decided to screen the flavodoxin to we screened flavodoxin
- P10 Line 5: remove most likely
- P10 Line 29: remove capital letter of fldX
- P12 Line 4: remove capital letter of fldX

Discussion

- P15 Line 11: add FldX gene number
- P15 Line 22: remove s from needs
- P15 Line 25: space missing after 2021
- P16 Line 2: wrong place for the verb
- P16 Line 4: idem
- P18 Line 12: add be after cannot ; change but to and
- P18 Line 15-16: check tenses and merge the two sentences
- P18 Line 19-20: check
- P18 Line 26: stress to stresses
- P19 Line 7: remove Now to e.g. In this condition
- P19 Line 12: remove already
- P19 Line 15: remove thus
- P20 Line 13: remove mostly

Material and Methods

- P22 Line 1: BLAST capital letters
- P24 Line 10: RT-PCR, q missing

We are very grateful for the notes on spelling and grammar within the manuscript. We have corrected all noted errors.

Staff Comments:

Preparing Revision Guidelines

Please return the manuscript within 60 days; if you cannot complete the modification within this time period, please contact me. If you do not wish to modify the manuscript and prefer to submit it to another journal, please notify me of your decision immediately so that the manuscript may be formally withdrawn from consideration by Microbiology Spectrum.

Re: Spectrum01895-23R1 (Characterizing the Flavodoxin Landscape in *Clostridioides difficile*)

Dear Dr. Susanne Sievers:

Thank you submitting your manuscript to Spectrum.

Your revised manuscript has now been reviewed by both Reviewers. One of the reviewers have a few further minor points which I would like you to address after which I would be happy to accept this paper for publication.

Revision Guidelines

Sincerely,
Meera Unnikrishnan
Editor
Microbiology Spectrum

Reviewer #3 (Comments for the Author):

-Add in the supplementary table S4 the gene number corresponding to Fuchs et al. to facilitate the navigation through Clostbase.

-In table S4 I would remove the column about the operons in the Soutourina et al. part as they are not discussed as such in the article and the density of the data does not seem compatible with such interpretation.

-About the comparison of the transcriptomic data, I was actually thinking of the Weiss et al 2021 (Ref 43 in the manuscript) that

would be more useful that the Giordano's

-The half-life data on CD2825 mRNA is based solely on one replicate (instead of 3 for the other gene tested), I don't understand why.

-Authors should justify the use of codY as reference gene in RT-qPCR

Re: Spectrum01895-23R1 (Characterizing the Flavodoxin Landscape in *Clostridioides difficile*)

Dear Dr. Susanne Sievers:

Thank you submitting your manuscript to Spectrum.

Your revised manuscript has now been reviewed by both Reviewers. One of the reviewers have a few further minor points which I would like you to address after which I would be happy to accept this paper for publication.

Revision Guidelines

Publication Fees: For information on publication fees and which article types are subject to charges, visit our <https://journals.asm.org/publication-fees> target="blank">website. If your manuscript is accepted for publication and any fees apply, you will be contacted separately about payment during the production process; please follow the instructions in that e-mail. Arrangements for payment must be made before your article is published.

Sincerely,
Meera Unnikrishnan
Editor
Microbiology Spectrum

Reviewer #3 (Comments for the Author):

-Add in the supplementary table S4 the gene number corresponding to Fuchs et al. to facilitate the navigation through Clostrbase.

The reviewer is correct that the indication of the gene number corresponding to Fuchs et al. makes navigation through Clostrbase much easier. Therefore, we have supplemented the annotation under the GenBank accession number AM180355.1 with the annotations of the accession number CP010905.2 used by Fuchs et al.

-In table S4 I would remove the column about the operons in the Soutourina et al. part as they are not discussed as such in the article and the density of the data does not seem compatible with such interpretation.

The data of Soutourina *et al.* show TSS for *wrbA* and *fldX*, which were not found in the analysis of Fuchs et al. and thus complement the data. Since Table S4 is intended to provide a broad overview of the operon structure including the TSS and TTS of the flavodoxins in *C. difficile* 630, we would prefer to leave the data of Soutourina et al. in the table.

-About the comparison of the transcriptomic data, I was actually thinking of the Weiss et al 2021 (Ref 43 in the manuscript) that would be more useful than the Giordano's

We agree with the reviewer, that the Weiss *et al.* publication fits more to our transcription data, than Giordano's.

Weiss *et al.* were able to show in their transcriptome experiments that *fldX* is still the most strongly induced gene in *C. difficile* 630 even after 8 hours under microaerophilic conditions (1.5 % O₂). This information is very interesting and complements our data by showing that *fldX* expression increases very rapidly after oxidative stress exposure and remains strongly induced over time under microaerophilic conditions. We have included the information of the Weiss *et al.* publication on *fldX* expression in more detail in the manuscript.

-The half-life data on CD2825 mRNA is based solely on one replicate (instead of 3 for the other gene tested), I don't understand why.

We apologize that we could only show one biological replicate for the calculation of the RNA half-life of CD2825 in the first revision, although we examined three biological replicates. Due to technical issues with the Northern Blot analysis, RNA samples of two of the replicates were thawed and frozen several times. For these two replicates no signal for CD2825 RNA could be detected anymore confirming the instability of the RNA. As we wanted to meet the deadline for submission of the revised manuscript, we decided to show the data of the one replicate for CD2825 we had.

In the second revision, we repeated the half-life experiment, harvested new RNA and performed 2 additional replicates. Data on the short half-life of *CD2825* are now based on three biological replicates.

-Authors should justify the use of *codY* as reference gene in RT-qPCR

The following statement was added to the manuscript: "*codY* was used as a reference gene for normalization because its expression is stable under the different iron stress conditions, oxidative stress conditions and the combination of both stress conditions".

In preliminary experiments, a total of 8 different potential reference genes were tested across the different stress conditions. The genes tested included *rpoC*, *rrs*, *codY*, *gyrA*, *pol III C*, *tpiA*, *rpsJ*, *recA*. Among all tested genes, only *codY* showed stable expression across all tested conditions and was therefore the only one suitable as a reference gene for RT-qPCR.

Re: Spectrum01895-23R2 (Characterizing the Flavodoxin Landscape in *Clostridioides difficile*)

Dear Dr. Susanne Sievers:

Your manuscript has been accepted, and I am forwarding it to the ASM production staff for publication. Your paper will first be checked to make sure all elements meet the technical requirements. ASM staff will contact you if anything needs to be revised before copyediting and production can begin. Otherwise, you will be notified when your proofs are ready to be viewed.

Sincerely,
Meera Unnikrishnan
Editor
Microbiology Spectrum